



**Six-year source apportionment of submicron organic aerosols from near-**
**continuous measurements at SIRTA (Paris area, France)**
Yunjiang Zhang[1,2*], Olivier Favez[1*], Jean-Eudes Petit[2], Francesco Canonaco[3], Francois Truong[2], Nicolas
Bonnaire[2], Vincent Crenn[2†], Tanguy Amodeo[1], Andre S.H. Prévôt[3], Jean Sciare[2,4], Valerie Gros[2],
Alexandre Albinet[1]
[1]Institut National de l'Environnement Industriel et des Risques, Verneuil-en-Halatte, France
[2]Laboratoire des Sciences du Climat et de l'Environnement, CNRS-CEA-UVSQ, IPSL, Université
Paris-Saclay, Gif-sur-Yvette, France
[3]Laboratory of Atmospheric Chemistry, Paul Scherrer Institute, Villigen PSI, Switzerland
[4]Energy, Environment Water Research Centre, The Cyprus Institute, Nicosia, Cyprus
[†]Now at ADDAIR, Buc, France
* Corresponding authors: yjanzhang@gmail.com and olivier.favez@ineris.fr
**Abstract**
Organic aerosol (OA) particles are recognized as key factors influencing air quality and climate
change. However, highly-time resolved year-round characterizations of their composition and
sources in ambient air are still very limited due to challenging continuous observations. Here,
we present an analysis of long-term variability of submicron OA using the combination of
Aerosol Chemical Speciation Monitor (ACSM) and multi-wavelength aethalometer from
November 2011 to March 2018 at a background site of the Paris region (France). Source
apportionment of OA was achieved via partially constrained positive matrix factorization (PMF)
using the multilinear engine (ME-2). Two primary OA (POA) and two oxygenated OA (OOA)
factors were identified and quantified over the entire studied period. POA factors were
designated as hydrocarbon-like OA (HOA) and biomass burning OA (BBOA). The latter factor
presented a significant seasonality with higher concentrations in winter with significant



monthly contributions to OA (18-33%) due to enhanced residential wood burning emissions.
HOA mainly originated from traffic emissions but was also influenced by biomass burning in
cold periods. OOA factors were distinguished between their less- and more-oxidized fractions
(LO-OOA and MO-OOA, respectively). These factors presented distinct seasonal patterns,
associated with different atmospheric formation pathways. A pronounced increase of LO-OOA
concentrations and contributions (50-66%) was observed in summer, which may be mainly
explained by secondary OA (SOA) formation processes involving biogenic gaseous precursors.
Conversely high concentrations and OA contributions (32-62%) of MO-OOA during winter and
spring seasons were partly associated with anthropogenic emissions and/or long-range
transport from northeastern Europe. The contribution of the different OA factors as a function
of OA mass loading highlighted the dominant roles of POA during pollution episodes in fall and
winter, and of SOA for highest springtime and summertime OA concentrations. Finally, long-
term trend analyses indicated a decreasing feature (of about 200 ng m$^{-3}$ yr$^{-1}$) for MO-OOA,
very limited or insignificant decreasing trends for primary anthropogenic carbonaceous
aerosols (BBOA and HOA, along with the fossil fuel and biomass burning black carbon
components), and no trend for LO-OOA over the 6$^{+}$-year investigated period.



## 1 Introduction


Organic aerosol (OA) particles account for a large mass fraction of submicron aerosol (PM$_1$) in
the atmosphere (Zhang et al., 2007) and play a key role in regional air pollution and climate
(Boucher et al., 2013). OA originates from i) primary emission sources (primary OA, POA),
directly emitted into the atmosphere by anthropogenic activities (e.g., fossil-fuel and biomass
combustions) or biogenic emissions (e.g., pollen, bacteria, fungal and fern spores, viruses, and
fragments of plants), and ii) secondary formation via atmospheric oxidation processes of gas
precursors, i.e., biogenic and anthropogenic volatile or semi-volatile organic compounds
(VOCs or SVOCs) (Kroll and Seinfeld, 2008; Hallquist et al., 2009; Nozière et al., 2015). Due to
their multiplicity and complexity, these various sources and physicochemical mechanisms
remain poorly documented and understood. Although numerous time-limited field campaigns
allowed to greatly improve our knowledge of OA properties in the last decade (e.g., Jimenez
et al., 2009; Lanz et al., 2010; Zhang et al., 2011; Shrivastava et al., 2017; Li et al., 2017;
Srivastava et al., 2018a, and references therein), similar studies performed on a multi-year
scale remain scarce and particularly challenging (Fröhlich et al., 2015a; Schlag et al., 2016; Sun
et al., 2018). Long-term observations and source apportionment of OA are nevertheless
necessary to better quantify the contribution of airborne OA particles to air quality and to set-
up scientifically-sound emission control strategies. They can also contribute to a better
understanding of the atmospheric fate of OA and reduce uncertainties associated with its
(in)direct radiative forcing.
Online aerosol characterization techniques, such as aerosol mass spectrometry (AMS),
have demonstrated their capacity to improve our knowledge of key aerosol chemical
components – such as OA - by providing highly time-resolved mass spectral data for the
nonrefractory PM$_1$ fraction (NR-PM$_1$) (Jayne et al., 2000; Canagaratna et al., 2007). Using
receptor model approaches, especially positive matrix factorization (PMF) (Paatero and
Tapper, 1994), OA measured by AMS techniques can be further portioned into various source
factors using statistic models (Ulbrich et al., 2009; Zhang et al., 2011). For example,
hydrocarbon-like OA (HOA) is frequently identified within urban environments and attributed
to primary emissions from fuel consumption (Zhang et al., 2007; Jimenez et al., 2009), while
biomass burning OA (BBOA) is often resolved specifically during cold seasons or within wild
fire plumes (Alfarra et al., 2007; Lanz et al., 2010; Zhou et al., 2017). Oxygenated OA (OOA),



commonly considered as a surrogate for SOA, is ubiquitously observed in urban, suburban and
remote environments (Zhang et al., 2007; Srivastava et al., 2018a; Zhang et al., 2011; Crippa
et al., 2014). OOA can be further separated into different fractions, being for instance
classified according to its atmospheric ageing (OOA-type 1 vs. OOA-type 2) or alternatively
described as more oxidized (MO-OOA) or less oxidized (LO-OOA) compared to each other
(Jimenez et al., 2009; Ng et al., 2011a; Sun et al., 2018). Different OOA factors can also be
identified as relevant to various sources of SOA precursors, such as anthropogenic activities
(e.g., traffic and biomass burning emissions) (Gilardoni et al., 2016; Gentner et al., 2017) and
biogenic emissions (e.g., isoprene and monoterpenes) (Xu et al., 2015; Zhang et al., 2018;
Freney et al., 2018) in specific regions and/or seasons. Such source apportionment has the
potential to support air quality strategies and shall be able to assess the efficiency of
mitigation measures of emission pollutants.
Over the last decades, particulate matter (PM) and anthropogenic VOCs emissions in
Europe have been drastically reduced in many activity sectors by stringent emission
regulations (EMEP, 2016). However, their impacts on both ambient POA and SOA
concentrations are poorly assessed and still suffer from the lack of long-term observational
data. Based on a less-advanced but more robust technology than AMS, the aerosol chemical
speciation monitor (ACSM) has been designed to provide continuous measurements of the
main non-refractory chemical species within submicron aerosols (Ng et al., 2011b). As for the
AMS, OA mass spectra obtained by the ACSM can be used in PMF analysis for quantification
of OA sources (e.g., Sun et al., 2012; Fröhlich et al., 2015b; Zhang et al., 2015). So far, several
long-term OA source apportionment studies have been reported based on ACSM
measurements at various sites (Canonaco et al., 2015; Fröhlich et al., 2015a; Schlag et al., 2016;
Reyes-Villegas et al., 2016; Rattanavaraha et al., 2017; Sun et al., 2018). However, these
studies have been limited to periods up to 2-years durations.
The longest ACSM timeseries recorded so far (from end of 2011 onwards) is used here
to investigate OA sources at a regional background site of the Paris region (France), which is
one of the largest urbanized regions in Europe. It has already been demonstrated that OA
plays a dominant role in controlling atmospheric pollution in this region (Bressi et al., 2013;
Petit et al., 2015). Furthermore, time-limited (typically, 1–2 months) measurement campaigns
demonstrated that primary fine aerosols are mainly influenced there by traffic emissions all



over the year and residential wood burning during cold seasons, while secondary aerosols
originate from both local production and regional transports (Sciare et al., 2011; Crippa et al.,
2013a, Crippa et al., 2013b, Petit et al., 2014; Srivastava et al., 2018b). In the present study,
main OA factors were identified and quantified from 25 successive and seasonal PMF analyses
over 6⁺ years, with the objective of keeping consistency between these factors from one PMF
analysis to another. In this respect, sporadic and/or minor OA sources were not accounted in
this study. The seasonal variations, weekly and diel cycles, as well as the long-term temporal
trends of the major OA factors were investigated. The relative contributions of the various
POA and SOA fractions were also plotted as a function of total submicron OA loadings with
the objective to better identify the main OA sources responsible for regional pollution
episodes. Finally, the geographical origins of high loadings of SOA factors were investigated
using air mass back-trajectory analyses.



## 2 Sampling site and instrumentation

Long-term submicron aerosol on-line measurements used in this study were performed from 1st of November 2011 to 26th of March 2018 at the SIRTA facility (Site Instrumental de Recherche par Télédétection Atmosphérique, 2.15 ˚E, 48.71 ˚N; http://sirta.ipsl.fr/). This exploratory platform is part of the European Aerosols, Clouds, Trace gases Research InfraStructure (ACTRIS, www.actris.eu) (e.g., Pandolfi et al., 2018). It is located 25 km southwest of Paris city center and is considered as representative of the background air quality of the Paris region (Haeffelin et al., 2005; Petit et al., 2015).

Major submicron aerosol chemical species i.e., OA, nitrate, sulfate, ammonium, and chloride, were measured using quadrupole ACSM. These measurements were achieved continuously, always using the same instrument, during the investigated period. Data was missing only for a few periods corresponding to two field campaigns performed elsewhere (in fall 2012 and March 2013) and to few technical breakdown and maintenance periods. Detailed descriptions of the ACSM measurement principles and basic data analysis are given by Ng et al. (2011b). Briefly, fine aerosols are sampled into the ACSM system through a 100 mm diameter critical orifice mounted at the inlet of the $PM_1$ aerodynamic lens (Liu et al., 2007; Ng et al., 2011b). Then, submicron aerosol particles are impacted and vaporized at the temperature ($T$) of about 600 ˚C and detected using electron impact (70 eV) ionization mass spectrometry. The ACSM was operated at a time resolution of about 30 min with a scan rate of 0.2 s $amu^{-1}$ from $m/z$ 12 to 150 amu (atomic mass unit). Coarse particles were removed upstream using an URG cyclone separator (with the size cut-off diameter of 2.5 μm). Calibrations of the detector response factor were performed regularly (typically every 6 months) using ammonium nitrate solutions (Ng et al., 2011b; Freney et al., 2019). The 1.4 default value was used for the OA relative ion efficiency for the whole dataset (Canagaratna et al., 2007). The accuracy of these ACSM measurements and the overall good working conditions of the instrument were verified through the participation to the ACTRIS ACSM intercomparison exercises that took place at SIRTA in November - December 2013 and March - April 2016 (Crenn et al., 2015; Fröhlich et al., 2015b, Freney et al., 2019).

Co-located multi-wavelength aethalometer (Magee Scientific) datasets were also available for the purpose of the study, providing complementary information on equivalent Black Carbon (eBC) concentrations and sources. Two aethalometers were used successively:





from November 2011 to February 2013 (AE31 model) and then from March 2013 to March
2018 (AE33 model). Both instruments measure aerosol light attenuation at seven wavelengths,
i.e., 370, 470, 520, 590, 660, 880 and 950 nm. The detailed descriptions of the AE31 operation
at SIRTA and aethalometer data analysis can be found in Petit et al. (2015). The AE33 is an
advanced aethalometer version, which allows better assessment and compensation of the
filter-loading effect using two simultaneous light attenuation measurements performed at
different rates of particle accumulation onto the filter tape (Drinovec et al., 2015; Drinovec et
al., 2017). The mass concentration of equivalent black carbon (eBC) was estimated from
attenuation measurement performed at 880 nm as described by Petit et al. (2015) and Zhang
et al. (2018). A correction factor of 1.64 was applied to raw absorption data delivered by the
instrument as recommended within the ACTRIS network (Zanatta et al., 2016). Furthermore,
eBC could be discriminated between its two main combustion sources, i.e., fossil-fuel
combustion (eBC$_{ff}$) and wood burning emissions (eBC$_{wb}$) using the aethalometer model
(Sandradewi et al., 2008; Favez et al., 2010; Sciare et al., 2011; Drinovec et al., 2015). For these
calculations, eBC$_{ff}$ and eBC$_{wb}$ were associated with absorption Angström exponents - in the
wavelength range 470-950 nm - of 0.9 and 1.7, respectively. These values are also in
agreement with a recent study by Zotter et al. (2017).

In addition to ACSM and AE33 measurements, co-located off-line analyses were

performed from daily (24 h) PM$_{2.5}$ filter samples, collected and analyzed for their content in
Elemental and Organic Carbon (OC and EC, respectively) following the ACTRIS
recommendations (Zanatta et al., 2017; Zhang et al., 2018). Briefly, filters were collected using
a low volume sampler (Partisol Model 2025; Thermo Scientific) equipped upstream with a VOC
denuder system. Mass concentrations of OC and EC from August 2012 to March 2018 were
then quantified using a Sunset Lab OC/EC analyzer implemented with the EUSAAR-2 thermal-
optical protocol (Cavalli et al., 2010). As shown on Figure S1, good agreements were obtained
between eBC and EC measurements ($r^2$ = 0.79, slope = 0.94; N=1185 as well as between OA
and OC measurements ($r^2$ = 0.68). The slope of 2.14 obtained between submicron OA
measured by the ACSM and PM$_{2.5}$ OC filter-based measurements corresponded to the higher
range of values generally observed at (sub)urban background sites - typically 1.6-2.2 (e.g., Bae
et al., 2006; Aiken et al., 2008; Favez et al., 2010; Sun et al., 2011; Canagaratna et al., 2015
and references therein) - and may be partly explained by the fact that the filter sampling set-



up has been designed to minimize positive sampling artefacts but do not prevent from
negative ones. Results obtained from these comparisons with filter-based measurements
supported the validity of the datasets used in the present study.

Co-located measurements of nitric oxide (NO) and nitrogen dioxide ($NO_2$) were

performed with a $NO_2$/NO/NOx analyzer (model T200UP, Teledyne API, USA). Data
measurements were used for further constrain traffic related OA sources. The meteorological
parameters, including meteorological parameters including temperature ($T$), relative humidity
(RH), wind speed (WS), boundary layer height (BLH), and precipitation were obtained from the
main SIRTA ground-based meteorological station, (located at about 4 km North-East of the
aerosol monitoring site).

**3. Atmospheric data treatment procedures**
**3.1 PMF analysis**

Positive Matrix Factorization (PMF) algorithm is a bilinear receptor model (Paatero and

Tapper, 1994) which has been widely used in source apportionment of ambient OA measured
by AMS or ACSM (e.g., Ulbrich et al., 2009; Zhang et al., 2011; Crippa et al., 2014; Li et al.,
2017). As expressed in Eq. (1), observed OA mass spectral matrix (m/z-based $x_{ij}$, dimensions:
m × n) can be discriminated into several variables:

$$x_{ij} = \sum_{k=1}^{p} \left( g_{ik} \cdot f_{kj} \right) + e_{ij} \qquad (1)$$

where $g_{ik}$ and $f_{kj}$ refer to factor (source) timeseries and mass spectra profiles, respectively, and
$e_{ij}$ correspond to residuals that could not be fitted by the PMF model. In this equation, $i$ and $j$
refer to row (timely resolved ACSM measurement data point) and column (m/z) indices in the
organic matrix, respectively, while $p$ indicates the number of factors in the PMF solution.
Based on a least-squares algorithm, PMF algorithm aims to iteratively minimize residuals and
a fit parameter Q, defined in Eq. (2):

$$Q = \sum_{i=1}^{m} \sum_{j=1}^{n} (e_{ij}/\sigma_{ij})^2 \qquad (2)$$


where $\sigma_{ij}$ is the estimated uncertainty of each $m/z$ ($j$) concentration at each time-step ($i$) in the
so-called error matrix. Organic concentration and error matrices (with $m/z$ ranging from 13 to
100) were exported from the ACSM Local software (v 1.5.11.2). Downweighting of the $m/z$ 44-
group ions for the PMF model analysis was performed following procedures implemented in
the ACSM Local software and following data treatment strategy proposed by Ulbrich et al.

(2009).

When using PMF, it may be difficult to distinguish between factors with similar spectral

profiles, especially for ACSM datasets, which are associated with larger uncertainties
compared to AMS (Sun et al., 2012; Zhang et al., 2015; Fröhlich et al., 2015b). The source
finder (SoFi) toolkit, implemented with the ME-2 solver (Paatero, 1999), has recently been
developed by Canonaco et al. (2013) to better address this limitation. SoFi provides robust
functions which allow to constrain chosen factor profiles and/or timeseries. In particular, the
so-called $a$-value approach makes use of range-defining scalar values (with $a$ values ranging
from 0 to 1) in order to better elucidate specific PMF factor(s) profile(s) with a chosen degree
of freedom; the highest the $a$-value the less constrained the OA profile (Canonaco et al., 2013).
In the present work, this $a$-value approach has been used to constrain profiles of POA factors.
Some previous studies have already been performed at SIRTA using high resolution time-of-
flight AMS (HR-ToF-AMS) along with PMF analysis during short-time campaigns (typically
around 3-4 weeks), leading to the identification of HOA, BBOA, as well as a cooking OA (COA)
factor (Crippa et al., 2013a; Crippa et al., 2013b; Fröhlich et al., 2015b). Mass spectra obtained
from these studies were used here as references to constrain POA factors, because of the prior
know source information as constraints. Conversely, mass spectral profiles of possible OOA
factors were left unconstrained. It should be noted that Crippa et al. (2013c) resolved up to 3
different type of OOA factors and/or a marine OA (MOA) factor when combining HR-ToF-AMS
and proton-transfer-reaction mass spectrometer (PTR-MS) datasets obtained during a
summer and a winter campaign at SIRTA.

OOA factor profiles may differ with time, notably due to seasonal variations of a

several parameters such as meteorological conditions, photochemistry, atmospheric lifetime,
air masses origin, and/or of gaseous precursor origins. In order to better account for such
variability, individual PMF analyses were performed on a 3-month basis, i.e., winter
(December-January-February), spring (March-April-May), summer (June-July-August), and fall



(September-October-November), with a total number of 25 different PMF runs (7 for winters
and 6 for each of the other seasons). November 2011 and March 2018 data were included in
the winter 2011-2012 and winter 2017-2018 analyses, respectively.

To evaluate the influence of the chosen temporal PMF window (i.e., time duration of

data used in ME-2 runs) on the seasonal ME-2 model results, different timeframes (i.e., 15, 30,
60 and 90 days) were tested. As shown in Figure S2 (with winter 2017 data as an example),
the excellent consistency of those results from different scenarios suggest very limited
influence of PMF windows on determining the outputs of ME-2 analyses. To better assess the
variations in primary and secondary OA in different seasons over the 6[+]-years period and to
allow for some degrees of freedom within the model runs, the main OA factors, including both
POA factors (HOA and BBOA) and two SOA factors (a less oxidized OOA (LO-OOA) and a more
oxidized OOA (MO-OOA)), were calculated as the average of 50 convergent ME-2 runs with a-
values varying from 0 to 0.4. Moreover, results obtained with an a-value of 0.2 were also
compared to these results for sensitivity analyses (Fig. S3). The diagnostics of the final OA-
factor solution are further discussed in section 4.1.

## 3.2 Influence of biogenic SOA

Biogenic SOA (BSOA) might have a significant influence on OA loadings in mid-latitude regions
during summertime and be further apportioned using AMS techniques (e.g., Leaitch et al.,
2011; Canonaco et al., 2015). For that reason, influence of this biogenic OA source was
specifically investigated in the present study. To do so, BSOA derived from terpene emissions
($BSOA_t$) was taken as a surrogate for total BSOA and the temperature ($T$) dependence of the
$BSOA_t$ formation process yield during summertime was simulated using a simple terpene
emission model (Goldstein et al., 2009; Schurgers et al., 2009; Leaitch et al., 2011 and
references therein), where an exponential curve function is describing the relation between
terpene emission rate ($\gamma$) and the air $T$, following Eq. 3:

$$\gamma = \gamma_0 \times e^{\beta(T-303)} \quad (3)$$

where $\gamma_0$ stands for the emission rate ($\mu g\ g^{-1}\ h^{-1}$) at standard conditions, and β is an empirical
constant chosen here to be equal to 0.09 $K^{-1}$ (Schurgers et al., 2009; Leaitch et al., 2011). As





reported by previous studies, biogenic terpene emissions could be a major source of such PMF
LO-OOA factor observed during summertime in western Europe (e.g., Canonaco et al., 2015;
Daellenbach et al., 2017; Daellenbach et al., 2019). Given that, $BSOA_t$ was assumed to be
mainly included in the LO-OOA fraction in the present work, and $BSOA_t$ estimated
concentrations were compared to LO-OOA concentrations data points corresponding to the
daytime maximum $T$ (at approximately 16:00 – 17:00 local time) in summer. Assuming that
LO-OOA could actually be mostly composed of $BSOA_t$ during this period of the day and
following the procedure described by Leaitch et al. (2011), the daily mass concentrations of
$BSOA_t$ were estimated as follows:
$$BSOA_{t,estimated} = LO - OOA_{(Observed\ at\ Tmin)} \times \frac{\gamma}{\gamma_{(Tmin)}} \qquad (4)$$
where $T_{min}$ corresponds to the lowest daily maximum $T$ observed across the investigated
summer seasons (i.e., 12°C ± 1°C) and LO-OOA$_{(observed\ at\ Tmin)}$ corresponds to the mean LO-OOA
concentration obtained for these data points (0.7 ± 0.3 µg m$^{-3}$, N = 17).

**3.3 Trend analysis**

The multi-year trends of OA factors obtained from the ME-2 analysis were analyzed using the
Mann-Kendall (MK) trend test. The MK test is a nonparametric test for monotonic trend in a
timeseries (Mann, 1945). The MK test is better suitable for nonnormally distributed, censored
and missing data, compared to parametric statistical tests, such as the $t$ test. The normality of
the mass concentrations of the OA factors was examined by the Shapiro-Wilk normality test
(Shapiro and Wilk, 1965). As a result of the Shapiro-Wilk normality test, all datasets of the
mass concentrations of the four OA factors were not normally distributed cases. The MK test
associated with Sen's estimator of slope (Sen, 1968) is insensitive to outliers, while it is not
appropriate for the chosen dataset with significant seasonality. Thus, the seasonal MK test
was used for the trend analysis when observed data had a significant seasonality with the
Kruskal-Wallis test (Kruskal and Wallis, 1952). The trend computation was performed here
using a R trend package (Pohlert, 2018). We applied monthly average data for all those tests
to illustrate the smoothed structure.


### 3.4 Air mass back-trajectory analysis

The HYbrid Single Particle Lagrangian Integrated Trajectory model (Hysplit) Draxler and Rolph, 2003; Stein et al., 2015) was applied to calculate 72-h back trajectories hourly arriving at SIRTA at a height of 100 m above ground level, based on GDAS meteorological data. The potential source contribution function model (PSCF) (Polissar et al., 1999) was used in this study to investigate the potential source origins that may contribute to high concentrations of OA factors at SIRTA. This analysis was achieved with a resolution of 0.2° × 0.2° for each grid cell, using the ZeFir toolkit (Petit et al., 2017). The probability function for a given grid cell ($i, j$), where $i$ stands for the latitude and $j$ for the longitude, is related to observed concentrations that are higher than a threshold value, which is defined by Eq. (6):

$$PSCF_{(i,j)} = \left(\frac{m_{ij}}{n_{ij}}\right) \cdot w_{ij} \tag{6}$$

where $m_{ij}$ is the total number of selected trajectory endpoints (i, j) associated with receptor concentrations of PMF factors higher than the threshold value, and $n_{ij}$ is the total number of back trajectory endpoints at each grid cell (i, j). The 75[th] percentile of each OA factors during the entire study was used as the threshold value to calculate $m_{ij}$. To reduce uncertainty caused by small $n_{ij}$ values for the PSCF modelling, an arbitrary weighting function ($w_{ij}$) was applied using Eq. (7) (Waked et al., 2014). To minimize the influence of some trajectories on the possible pathways of air mass transport, observed data points associated with low wind speed conditions (WS < 4 m s$^{-1}$) were filtered out. In addition, observed data points at SIRTA during the period with any hourly precipitation events (precipitation > 0 mm) were removed to reduce influence of wet deposition on ambient aerosol concentrations.

$$w_{ij} = \begin{cases} 1 \ for \ \log(n_{ij} + 1) \ \geq 0.85 \max \log(n_{ij} + 1) \\ 0.725 \ for \ 0.6 \ \max \log(n_{ij} + 1) \leq \log(n_{ij} + 1) < 0.85 \ \max \log(n_{ij} + 1) \\ 0.475 \ for \ 0.35 \ \max \log(n_{ij} + 1) \leq \log(n_{ij} + 1) < 0.6 \ \max \log(n_{ij} + 1) \\ 0.175 \ for \ \log(n_{ij} + 1) < 0.35 \ \max \log(n_{ij} + 1) \end{cases} \tag{7}$$

## 4 Results and discussion

### 4.1 Identification of the main OA factors





**4.1.1 Determination of the optimum factor number**
The optimal number of PMF OA factors shall be determined by the distribution of the
main sources at a given sampling site. Based on results obtained from the compilation of
previous AMS studies reported in the Paris region, two POA factors - HOA and BBOA - and two
OOA fractions - MO-OOA and LO-OOA – are undoubtedly major fraction of submicron aerosols
in Paris area over the year (Crippa et al., 2013a; Crippa et al., 2013b; Freutel et al., 2013; Petit
et al., 2014; Fröhlich et al., 2015b). Another POA source, i.e., COA, has also been identified
using HR-ToF-AMS during previous campaigns in Paris region (Crippa et al., 2013a; Crippa et
al., 2013b; Fröhlich et al., 2015b). However, the distinction between COA and HOA factors
based solely on ACSM measurements remains challenging due to highly similar mass spectra
and uncertainties associated with the ACSM low mass spectral resolution (Petit et al., 2014;
Fröhlich et al., 2015b).
To better assess a potential role of COA in our source apportionment study, several
ME-2 runs were conducted constraining either three POA factors (HOA, BBOA, COA) or two
(HOA, BBOA). In these tests, POA reference mass spectra determined by Fröhlich et al. (2015b)
were employed as anchor profiles (with $a$-values ranging from 0 to 0.4 with steps of 0.05).
PMF solutions with a factor number ranging from 3 to 6 were investigated on ACSM datasets
corresponding to different seasons of different years (December 2011 - February 2012, March
- May 2015, June - August 2017, September - November. 2017, December 2017 - February
2018). Results obtained from these preliminary individual PMF runs showed very good
consistency between them with two unconstrained OOA factors - MO-OOA and LO-OOA -
always appearing in the 4-factor (with constrained HOA and BBOA factor) and 5-factor (with
constrained HOA, BBOA and COA factor) solutions. Conversely, 3- and 6-factor PMF analyses
generally led to unsatisfactory solutions.
Figures 1 and S4 present results obtained for the 4- and 5-factor solutions, respectively,
for the winter 2017-2018 period, taken here as an example. In both cases, mass spectra were
in good agreement with those reported in the literature. However, the COA and BBOA factors
are displaying very similar diel patterns, leading to surprisingly good correlations between
these two factors (see Figure S5). It could then be concluded that COA-like aerosols at SIRTA
were primarily linked with wood burning emissions and pure cooking aerosols were probably
present in too low loadings to be properly quantified within the present study. This



assumption is consistent with conclusions drawn by other studies performed at SIRTA (Petit
et al., 2014; Srivastava et al., accepted for publication) as well as other studies showing that
the COA factor could not be solely attributed to cooking aerosols (e.g., Freutel et al., 2013,
Dall'Osto et al., 2015).

Therefore, the 4-factor solution, including two constrained POA factors (BBOA and

HOA) and two unconstrained factors, was chosen here as the "best estimate" for the PMF
runs performed over the long-term dataset. A total of 25 seasonal and individual PMF analyses
were then conducted using a similar procedure. The seasonal OOA factor mass spectra are
presented in Figure S6, showing high seasonal consistency for each OA factor. Moreover, as
shown in Figure S7, the distribution of residuals derived from the these 4-factor solution ME-
2 runs was sharply centered around 0, suggesting insignificance of possible unresolved OA
factor(s).

### 4.1.2 Source attribution

BBOA mass spectra are quite constant throughout the seasons, and present

characteristic peaks at m/z 29, 60, and 73 indicative of biomass burning combustion (Figure
S6). As shown on Figure 2a, BBOA diel cycles displayed well-marked patterns with strong
nighttime maxima, especially during the weekend. This confirms the predominance of
residential wood burning activities on BBOA concentrations at SIRTA and in the Paris region,
as already shown previously (e.g., Favez et al., 2009; Sciare et al., 2011; Crippa et al., 2013b;
Petit et al., 2014). As expected, BBOA diel cycles are similar to the ones obtained for $eBC_{wb}$,
except for small $eBC_{wb}$ morning peaks that were not observed for BBOA (possibly due to
uncertainties of the aethalometer model) and for lunch-time shouldering within BBOA
patterns, which might be related to limited COA emissions (see above). Interestingly, both of
this $eBC_{wb}$ and BBOA daytime rises were not observed during week-end, suggesting the
influence of local emissions related to working activities (e.g., eBC from commuting road
transport and staff canteens).

Compared to BBOA, HOA shows a more complex weekly diel pattern (Figure 2b). Its

pattern is generally similar to $eBC_{wb}$ and $NO_x$ (both being considered here as markers for traffic
emissions). HOA presents two peaks during working day, one in the morning and another in





the evening. Morning peaks, occurring during traffic rush hours are clearly indicative of road
transport contributions, confirming HOA as a proxy for traffic emissions. However, HOA
evening peaks occurs globally later than eBC$_{ff}$ and NO$_x$ ones (9:00-10:00 PM vs. 7:00 PM,
respectively) and much lower ratio are observed between HOA and eBC$_{ff}$ in the morning than
in the evening. This might be partly explained by i) higher eBC traffic emission factor in the
morning and/or ii) impacts of biomass burning sources on HOA concentrations in the late
evening. Moreover, eBC$_{ff}$ shows a clear weekend effect, with less-pronounced pattern on
Saturday and Sunday due to road transport reduction, while HOA displays intense nighttime
peaks during weekend. This HOA mean pattern was substantially influenced by winter data,
whereas summertime patterns display better consistency between HOA, eBC$_{ff}$ and NO$_x$ (Figure
S8). Altogether, these results claimed for considering HOA as a mixed factor partly composed
of traffic and biomass burning aerosols. This statement is in good agreement with conclusions
from complementary studies showing wood burning contribution to HOA at the same site
(Petit et al., 2014; Srivastava et al., accepted for publication). It was further supported by
higher m/z 44 contribution within HOA mass spectra in fall and winter than during the spring
and summer seasons (Figure S6), which could be characteristic of the presence of processed
biomass burning emissions (e.g., Grieshop et al., 2009; Fröhlich et al., 2015b).
As presented in Figures 1 and S6, MO-OOA mass spectra present a strong peak at *m/z*
44. In fact, this spectrum has been widely reported as low volatility OOA (LV-OOA) and
considered as composed of highly oxidized and aged SOA (Lanz et al., 2007; Ulbrich et al., 2009;
Zhang et al., 2011; Ng et al., 2011a). Compared to the poorly pronounced diel variability of
sulfate, this MO-OOA factor exhibits a slight enhancement at nighttime (Figure 2c), suggesting
a possible local formation mechanism involving nighttime chemistry, on top of its overall
regional feature. The geographic origins of the MO-OOA factor are further discussed in section
4.2.1 for each season.
The mass spectra of LO-OOA in this study present a higher *m/z* 43 and a lower *m/z* 44
(Figures 1a and S6), compared to MO-OOA, which is consistent with the mass spectral pattern
of previously reported freshly-formed semi-volatile OOA (SV-OOA) (Jimenez et al., 2009; Ng
et al., 2010). The diel variations of LO-OOA display higher concentrations during nighttime
than daytime (Figure 2d), with relative variations much more pronounced than for the MO-
OOA diel pattern. These results support different formation pathways of the two OOA





fractions. In winter, LO-OOA mass spectra has higher contributions of m/z 29 as well as
elevated m/z, i.e., starting from m/z 60, than during other seasons (Figure S6). Such
characteristics suggest a major influence of biomass burning emissions onto the LO-OOA
factor during wintertime, as previously proposed from measurements at SIRTA (e.g., Crippa et
al., 2013c). Conversely, in summer, this factor may be significantly influenced by BSOA
formation (Canonaco et al., 2015; Daellenbach et al., 2017). To investigate this possible origin,
we checked if summertime LO-OOA concentrations at higher daily $T$ were following
temperature dependence similar to the one expected for the formation of terpene SOA, as
explained in section 3.2. Results of these calculations are presented in Figure 3. LO-OOA
concentrations substantially increase with $T$, showing a good agreement with the estimated
$BSOA_t$ formation exponential profiles. However, when comparing with estimation derived
from Eq. (4) (referred to Figure 3), observed LO-OOA displays substantially higher loadings
than estimated $BSOA_t$ at highest concentration range. This could be partly due to the influence
of regional transports and atmospheric dilution on aerosol loadings and some possible
uncertainties (such as unclear formation schemes of biogenic SOA at SIRTA), which were not
considered in the $BSOA_t$ estimation. These comparison results between observation and
estimation indicates that the LO-OOA factor observed in summer might be mainly associated
with biogenic sources. This was aligned with the VOC seasonal patterns observed in the Paris
region (Baudic et al., 2016), although the underlying SOA formation mechanism is still unclear
and needs to be further investigated (Beekmann et al. 2015).

## 4.2 OA factor temporal variations

Figure 4 presents timeseries of total submicron OA and its four main factor components
(namely HOA, BBOA, MO-OOA and LO-OOA) together with key meteorological parameters:
boundary Layer height (BLH), relative humidity (RH) and temperature ($T$), during the entire
investigated period. Most meteorological parameters present seasonal cycles. For example,
the highest and lowest air $T$ were observed during summertime and wintertime, respectively,
while the highest RH was frequently observed in winter for each year. The highest BLH was
mainly observed in summer among all seasons. Total submicron OA presented dynamic
variations during all seasons with hourly average concentrations ranging from 0.03 to 77.5 μg
m$^{-3}$ and daily average values from 0.2 to 41.3 μg m$^{-3}$. There was no clear seasonality for the





total monthly average OA concentrations, varying from 4.8 to 5.1 µg m⁻³. However, each
individual OA factors displayed intra and inter-annual variations, which are discussed in this
section.

**4.2.1. Monthly and seasonal variations of OA factors**
Figure 5 illustrates monthly average concentrations obtained for each OA factor over the
studied period. HOA monthly concentrations vary from 0.4 to 1.3 µg m⁻³ and display a
statistically insignificant seasonal trend ($p > 0.05$, Figure 5a). Nevertheless, the mass
concentration of HOA is nearly twice higher during cold months (in the range of 0.9 – 1.3 µg
m⁻³, from November to March) than in other months (in the range of 0.4 – 0.5 µg m⁻³ from
April to October). This monthly cycle of HOA could be partially explained by lower BLH
conditions and influence of more intense emissions of biomass burning in cold seasons than
in warm seasons (Figures 4 and S9). As illustrated by Figure S10, HOA clearly presents two
peaks (in the morning and late evening) for each season. The evening HOA peak is significantly
higher than the morning peak in winter and fall seasons when high loadings of BBOA are
observed as well. Although dynamic processes (establishment of a stable nighttime boundary
layer) cannot be excluded, these results point to a possible contribution of biomass burning
emissions to the HOA factor in the evening during cold months, as discussed before from the
diel cycles of OA factors.

As shown in Figure 5b, BBOA displays a statistically significant seasonal pattern trend

($p < 0.0001$) with higher monthly mean concentrations (1.1 – 1.9 µg m⁻³) during cold months
(November – March) than during the April – September period (0.3 – 0.5 µg m⁻³). This seasonal
dependence of wood burning emissions is associated with the residential heating activities
over the Paris region. BBOA presents a seasonal dependence of its diel cycle, as presented in
Figure S10. In particular, BBOA shows an evident peak at evening/nighttime in winter, spring,
and fall, while it presents a stable diel cycle during summertime. The highest seasonally-
averaged nighttime peak (up to 2.4 µg m⁻³) is observed in winter, highlighting a significant
enhancement of wood burning emissions and influence of meteorological conditions (such as
low BLH) during this season.





Monthly average mass concentrations of MO-OOA present a significant seasonal trend
($p < 0.05$), varying from 1.0 in September to 3.5 µg m$^{-3}$ in March (Figure 4c), in agreement with
previous studies performed in Europe (Schlag et al., 2016; Daellenbach et al., 2017; Bozzetti
et al., 2017). The highest MO-OOA mass concentrations observed in the cold months are
somehow similar to the seasonal variation of BBOA. MO-OOA diel cycles also present a
seasonal variation, with significant increase during evening/nighttime in winter, spring, and
fall (Figure S10). In order to minimize the effect of atmospheric dilution and regional transport,
the mass concentration of MO-OOA was normalized to sulfate, the latter one being considered
as a regional secondary production marker (Petit et al., 2015 and Figure S11). As shown in
Figure S11, the correlations between MO-OOA and sulfate are found to be strongly BBOA- and
wind speed-dependent. For high wind speed and low BBOA concentrations, the mean MO-
OOA-to-sulfate ratio is close to 1, while it reaches up to 8 under high BBOA and low-to-medium
wind speed. This is consistent with the assumption of an enhancement of MO-OOA formation
in the presence of substantial biomass burning emissions, which have been reported as a
major anthropogenic SOA source (Heringa et al., 2011; Tiitta et al., 2016; Bertrand et al., 2017).
Furthermore, high concentrations of MO-OOA are generally observed at high RH (> 80 %) and
low $T$ (< 0 °C) conditions during wintertime (Figure S12) and the MO-OOA-to-sulfate ratio
shows a significant enhancement as a function of RH (Figure S13), suggesting that the
aqueous-phase heterogeneous processes may represent an important pathway for the local
MO-OOA formation in winter as proposed by Gilardoni et al. (2016). Conversely, there are no
obvious RH-$T$ dependent patterns for the MO-OOA in spring (Figure S12), indicative of more
complex formation processes during this season. In summer, MO-OOA displays evident
increase from early afternoon to evening (Figure S10), suggesting significant local
photochemical production of SOA particles in summer with higher $T$ and increased solar
radiation (Petit et al., 2015). As a matter of fact, MO-OOA presents high concentrations under
high $T$ (> 25 °C) and low RH (< 65%) summertime conditions (Figure S12). In conclusion, and
despite relatively constant mass spectra all over the year, MO-OOA appears to originate from
various seasonal-dependent formation pathways and sources (such as biomass burning and
biogenic sources), that should still be investigated in more details.
The LO-OOA mass spectra with high $f_{43}$ / $f_{44}$ ratios are frequently observed in spring,
summer and fall, whereas a lower ratio is obtained for winter (Figure S6). These different mass



spectra of LO-OOA could be partially explained by seasonal-dependent formation mechanisms
and sources. The monthly mean mass concentrations of LO-OOA vary from 0.8 to 3.6 µg m$^{-3}$
(Figure 5d) and shows a statistically significant seasonality ($p < 0.001$) with higher
concentrations during warm months and lower during cold months. As discussed above, the
highest summertime LO-OOA concentrations are assessed to be mainly linked with BSOA
formation. As presented in Figure S12, $T$-RH dependence of the LO-OOA factor is very different
according to the season. In particular, the highest wintertime LO-OOA concentrations are
mainly observed at low $T$ and high RH conditions, suggesting that gas-particle partitioning may
play an important role in LO-OOA formation during this season. In summer, the LO-OOA
concentrations present strong $T$ positive dependence while RH dependence is not clear,
indicating that photochemical production of LO-OOA became more important in summer than
in winter. Moreover, high concentrations of LO-OOA are observed at daytime in summer,
which is different from the diel variations in other seasons with high concentrations only
during nighttime (Figure S10). Such LO-OOA diel variations could further support the
photochemical processing dominating the LO-OOA production in summer.

**4.2.2. Long-term trends**

Figure 6 presents the results obtained from the trend analysis of the 6-year timeseries of the
four OA factors as well as eBC$_{ff}$ and eBC$_{wb}$ components. The significance and magnitude of
these trends were examined using the MK p-value and Sen's slope, respectively. BBOA
presents a statistically significant decreasing trend ($p < 0.05$) with a Sen's slope of about 80 ng
m$^{-3}$ per year in the Paris region. On the other hand, eBC$_{wb}$ concentration trends appear quite
stable over the investigated period. Two possible reasons may explain the discrepancy trend
results between BBOA and eBC$_{wb}$. It may be hypothesized that a limited overall improvement
of wood stove performances in the Paris region could have influence BBOA emission factors
more than eBC$_{wb}$ ones, but no evidence has been found to support this assumption. Similarly,
but in the opposite way, eBC$_{ff}$ was found to have a significant decreasing trend, while HOA
trend was found to be statistically insignificant ($p$-value $> 0.05$). However, if removing the high
concentration peak observed in December (for which an important contribution of wood
burning HOA can be expected), the MK $p$-value is reduced to be 0.024, which would be
indicative of a significant decreasing trend (with a related Sen's slope of 72 ng m$^{-3}$ per year).



These results would be in line with a reduction of PM traffic emissions over the past years in
France, as estimated by the French emission inventory state operator (CITEPA, 2018).
However, such trends analysis should be performed on longer datasets for a much better
evaluation of the pollution control strategies (both on road transport and residential heating
emissions) in the Paris region.
MO-OOA shows a significant decreasing trend ($p < 0.05$) with a Sen's slope of 204 ng
$m^{-3}$ per year. Considering the overwhelming secondary origin of this factor, this significant
decreasing trend may be partially explained by a reduction of anthropogenic VOCs emissions
in France over the investigated period (CITEPA, 2018). LO-OOA presents no significant trend
(with $p = 0.29$). As discussed above, higher LO-OOA loadings may be linked to BSOA formation,
especially at summertime. The stability of LO-OOA concentrations over time may be linked to
limited changes in biogenic VOC emissions and/or in relevant oxidant concentrations, that
control the SOA burden in the atmosphere. Effect of anthropogenic-biogenic interaction
mechanisms on biogenic SOA formation - e.g., involving $NO_x$, as reported by previous studies
in urban regions (Budisulistiorini et al., 2015; Zhang et al., 2017) - could also partially explain
the limited changes for the long-term trend of LO-OOA at SIRTA. Detailed LO-OOA formation
processes involved here still need to be further investigated. Nevertheless, it may be assumed
that reductions of anthropogenic VOC emissions only cannot be sufficient to weaken the total
SOA background concentrations in the Paris area.

### 4.3 OA source contribution as a function of OA concentrations

Figure 7 presents the contribution of the four OA sources as a function of total submicron OA
mass loadings or each season along with percent changes of meteorological conditions. In
winter and fall, all meteorological parameters - except limited changes in RH - show negative
relationships as a function of the OA mass concentrations, confirming the coincidence of low
$T$, low WS, and/or low BLH in the formation of pollution episodes (Dupont et al., 2016). POA
contributions gradually increase with increasing OA concentrations: an OA increase from
below 5 µg $m^{-3}$ to above 25 µg $m^{-3}$ leads to a POA contribution increase from 35 % (resp. 27 %)
up to 64 % (resp. 70 %) in winter (resp. fall). These results illustrate the major role of primary
sources during periods with high OA concentrations during the cold seasons. In particular,



BBOA contribution gradually increase from 21 % (15 %) to 41 % (40 %) in winter (fall) along
with OA mass loading increase.
In spring, OA composition is radically changed and is dominated by the two OOA
fractions, with almost constant average contributions $(68-77\%)$ regardless OA concentration
levels, indicating the major role of SOA during this season. MO-OOA presents higher
contributions to OA (45-53%) than LO-OOA (15-31%), suggesting that the formation of aged
SOA plays a key role on the build-up of episodes with high OA concentrations during
springtime. As shown in Figure 8f, the percent changes in $T$, WS, and BLH gradually decrease
with increasing OA concentrations. By contrast, RH shows a positive relationship with OA mass
concentrations, with the largest RH enhancement (16%) at highest OA-loading bin (> 25 μg m$^{-}$
$^{3}$). This may suggest that high RH being the most favorable environment condition for SOA
formation during springtime OA pollution episodes, as supported by a high contribution of
OOA factors at the highest OA concentration level (Figure 7b). In addition, although BBOA
contributions remained relatively limited, it increases from 11% to 17% when OA increased
from less than 10 μg m$^{-3}$ to > 25 μg m$^{-3}$. This may reveal a non-negligible influence of wood
burning emissions during early spring pollution episodes.
In summer, OA was also dominated by the two OOA fractions (around 80-85% at all
OA-loading bins). The LO-OOA contribution gradually increase from 51 % to 69% as a function
of OA mass loadings associated with a significant increase of $T$. Other meteorological variables
(i.e., RH, WS and BLH) showed relatively stable changes across different OA mass loadings
(Figure 7h). These results confirm that high OA concentrations during summer are strongly
determined by $T$-driven biogenic SOA formation processes.

**4.4 Potential geographic origins of SOA factors**
Figure 8 shows maps of the most probable geographic origins of the two OOA factors for each
season based on PSCF analysis. In winter, MO-OOA presents high PSCF values over the Benelux,
Germany and Poland, showing a major influence of long-range transport of OA from
northeastern sectors. Similar results are obtained from wind-dependent analyses (Figure S14).
This could be associated with more stable conditions with anticyclonic conditions, but could
also suggest more intense SOA production and aging processes at regional scale for



continental air masses. As a matter of fact, MO-OOA shows wider potential source regions
than LO-OOA, which is assessed as fresh SOA and could be mainly formed at more local scale
in winter. Moreover, the impact of transport from northeastern regions − hosting intense
industrial activities - onto MO-OOA concentrations may also support a significant
anthropogenic origin for this SOA factor.
As shown in Figures 8c-d and S14, both MO-OOA and LO-OOA present high springtime
PSCF values originating from the northeastern regions too, which can participate in pollution
episodes frequently observed during this season (Petit et al., 2015; Srivastava et al., 2018b).
Therefore, mitigation of VOCs emissions at the regional scale could help to reduce the
substantial influence of OA on PM limit value exceedances during this season.
Narrower distribution of potential source regions was observed in summer and fall,
compared to winter and spring. MO-OOA presents potential source regions mainly from the
northeast in summer (Figure 8e), while it has a high potential source region originating from
the south in fall (Figure 8g). Finally, summertime LO-OOA, possibly from biogenic sources,
presented potential source regions from both northeast and south, suggesting the
contribution of a regional transport to the biogenic SOA production in summer.
All these results indicated that significant reduction of the SOA burden in the Paris
region does not only require the limitation of local source emissions, but also needs a
synergistic control strategy for the regional sources, especially from northeastern European
regions. In this respect, they confirmed conclusions reached by previous short-term
campaigns (e.g., Sciare et al., 2010; Crippa et al., 2013b; Freutel et al., 2013; Beekman et al.,

2015).


## 620 5. Conclusions

A comprehensive OA source apportionment has been achieved over the region of Paris from
November 2011 to March 2018. 4 factors, comprising HOA, BBOA, MO-OOA and LO-OOA, have
been identified and selected to ensure consistency of PMF factor solution over 6 years in this
study. Mean annual contributions of these factors to OA were of 11-16 % (HOA), 14-19%
(BBOA), 25-42 % (LO-OOA), and 30-45 % (MO-OOA), respectively. BBOA presented a



statistically significant seasonal pattern with highest concentrations during cold months, due
to residential wood burning emissions. The contribution of BBOA increased with increasing
concentration of OA mass in winter and fall — along with decreasing boundary layer height and
wind speed — highlighting the importance of biomass burning emissions for OA pollution under
stagnant meteorological conditions. HOA presented temporal variations similar to BBOA in
cold months, which was partly related to the fact that wood burning emissions also
contributed to HOA burden. BBOA and HOA exhibited very limited (< - 0.1 µg m$^{-3}$ yr$^{-1}$) or not
significant trends during the 6$^+$-years investigated period. These results imply that specific
mitigation strategy, especially for residential wood burning, are still necessary for substantial
improvement of air quality in cold season in the Paris region.
LO-OOA and MO-OOA presented different seasonal variations, reflecting different
formation mechanisms and/or precursor sources. LO-OOA displayed a pronounced seasonal
cycle, with highest contribution total OA in summer (50-66 %) and lowest ones in winter (12-
19 %). Enhanced LO-OOA production during the warm season was assessed to be mainly
driven by biogenic SOA formation. This factor showed no significant long-term trend for the
studied period. MO-OOA presented higher contribution to OA at wintertime (35-51 %) and
springtime (32-62 %) than during the rest of the year. PSCF analyses suggested a high
probability of MO-OOA long-range transport from northeastern Europe towards the Paris
region. MO-OOA displayed a significant decreasing trend (of about 0.2 µg m$^{-3}$ yr$^{-1}$), which
might reflect the effect of emission control strategy of anthropogenic SOA precursors at the
regional scale over the last decade. However, future work is needed to fully understand
chemical properties of these SOA factors corresponding to different origins over different
seasons in the Paris region and to quantify the impact of emission control on ambient SOA
burden.

***Data availability.*** The data have been presented in the text and figures as well as supplement.
Additional-related data will be available upon request.

***Competing interests.*** The authors declare that they have no conflict of interest.



**Author contribution.** O.F., A.A., and V.G. designed and led the study. Y.Z. conducted the data analyses.
J-E.P., F.T., N.B., V.C., T.A., and J.S. provided the field observation. F. C. and A.P. supported the source
apportionment analyses. Y.Z. and O.F. interpreted the data, and wrote the manuscript, with inputs
from all coauthors.

**Acknowledgements.** This work has been part of the EU-FP7 and H2020 ACTRIS projects (grant
agreements no. 262254 and 654109) as well as the COLOSSAL COST action CA16109. It has also been
directly supported by the French Research Council (CNRS), the French alternatives energies and atomic
energy commission (CEA), and the French ministry of Environment through its funding to the reference
laboratory for air quality monitoring (LCSQA). Finally, Y. Zhang acknowledges the China Scholarship
Council (CSC) for PhD scholarship.





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

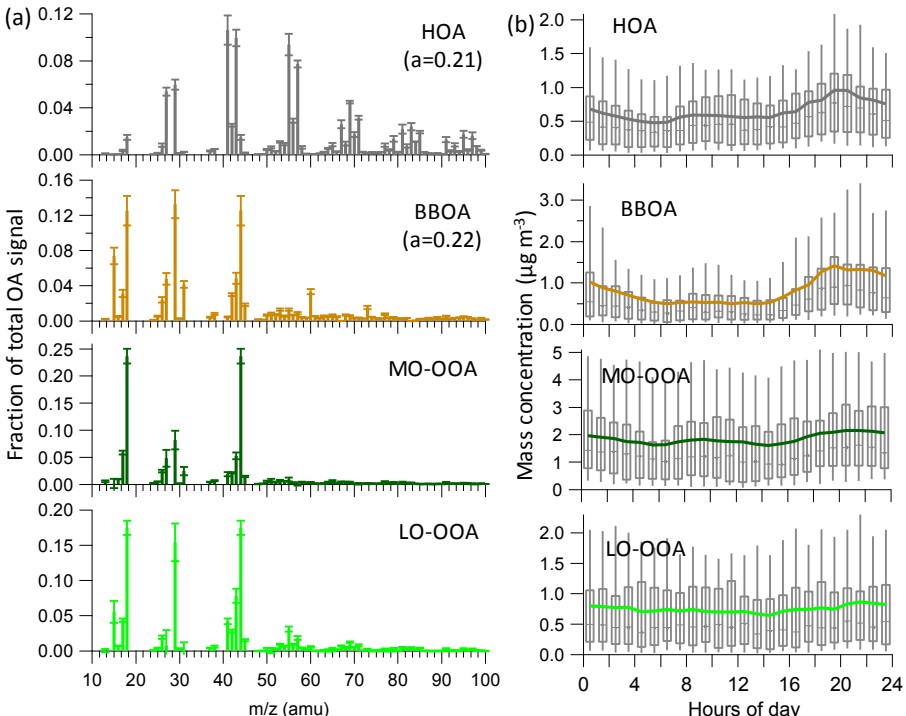

**Figure 1.** Mass spectra (a) and diel variations (b) of four OA factors obtained from the 4-factor solution of ME-2 runs for winter 2017-2018. In (a), error bars in each plot present 1 standard deviation. Stick lines indicate average values over all selected ME-2 runs. Averaged a-values for constrained factors during the ME-2 runs are also shown. In (b), the upper and lower boundaries of boxes indicate the 75th and 25th percentiles; the vertical lines within the box correspond to median values; the whiskers above and below boxes refer to 95th and 10th percentiles; and solid colored lines represent mean values.

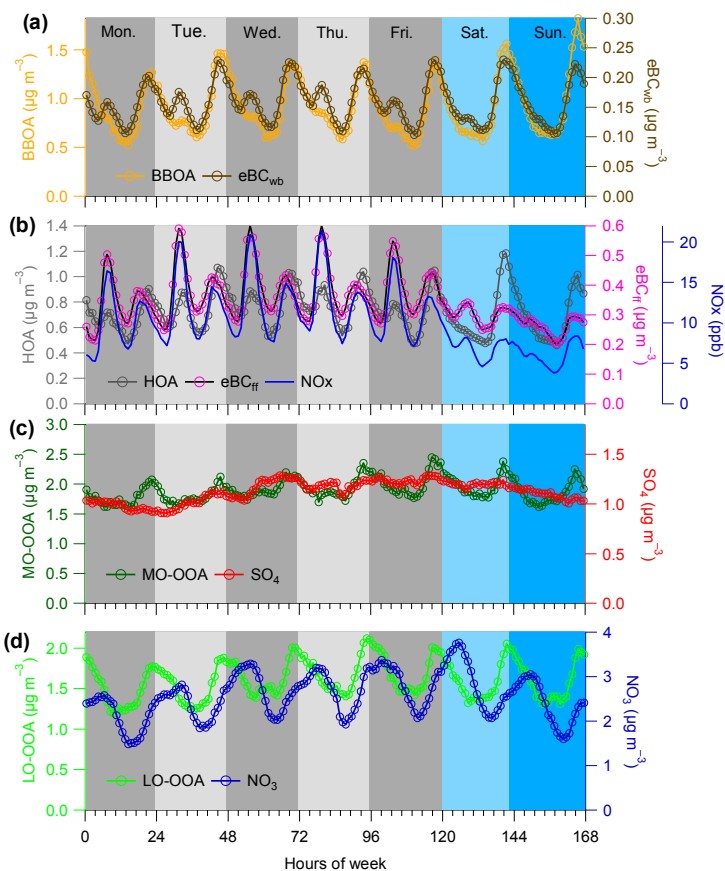


**Figure 2.** Weekly cycles averaged for the entire period of study for (a) HOA, (b) BBOA, (c) MO-OOA and (d) LO-OOA, along with possible external tracers (eBC$_{wb}$, eBC$_{ff}$ and NO$_x$, sulfate, and nitrate, respectively). Weekdays (24 h) are colored in different gray and weekend days in different blue.








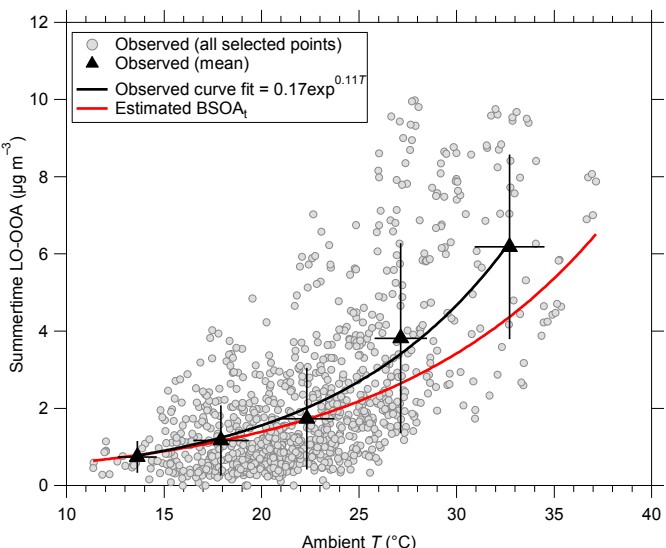


**Figure 3.** Temperature dependence of summertime LO-OOA obtained from observation and observationally constrained calculation based on biogenic terpene emissions model (Schurgers et al., 2009; Leaitch et al., 2011).




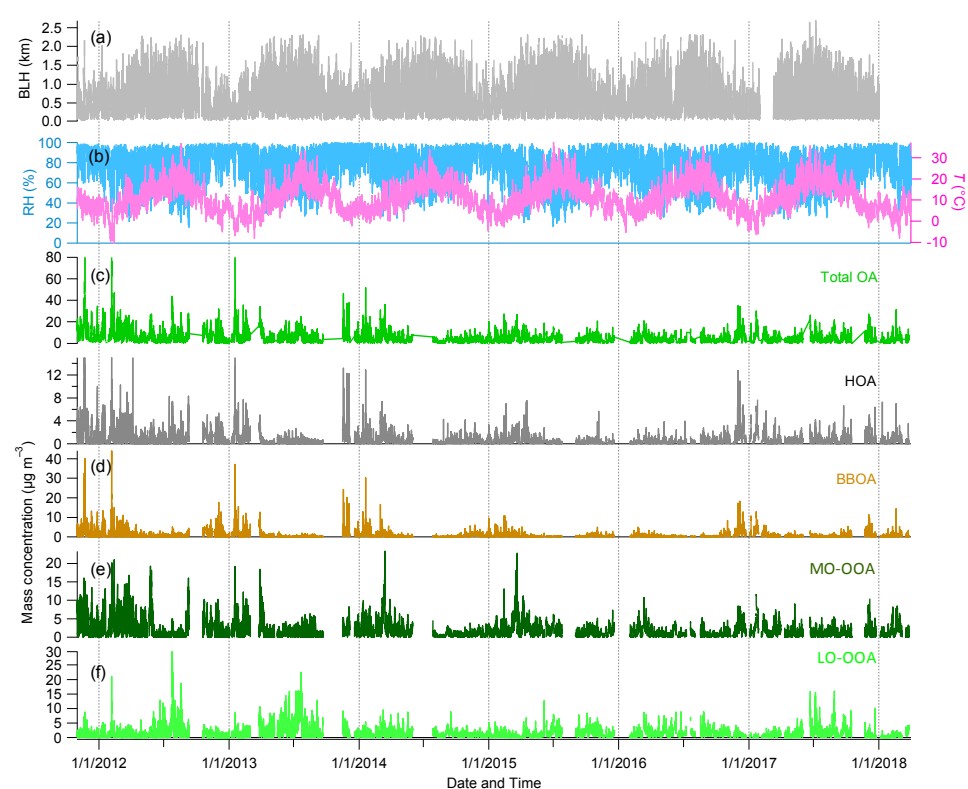


**Figure 4.** 6+-year timeseries of meteorological parameters, i.e., (a) boundary layer height (BLH);

and (b) relative humidity (RH) and temperature (*T*), and mass concentrations of (c) total OA

and four OA PMF factors, i.e., (c) HOA, (d) BBOA, (e) MO-OOA, and (f) LO-OOA.






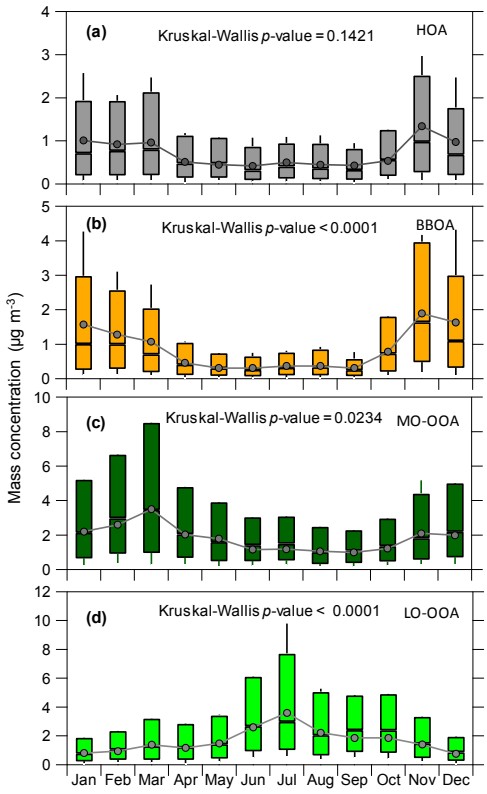

**Figure 5.** Monthly variations of the four OA factors and associated Kruskal-Wallis p-value for detecting seasonality. The box plots describe the different percentiles (10th, 25th, 50th, 75th, and 90th) and the mean (gray solid circle).



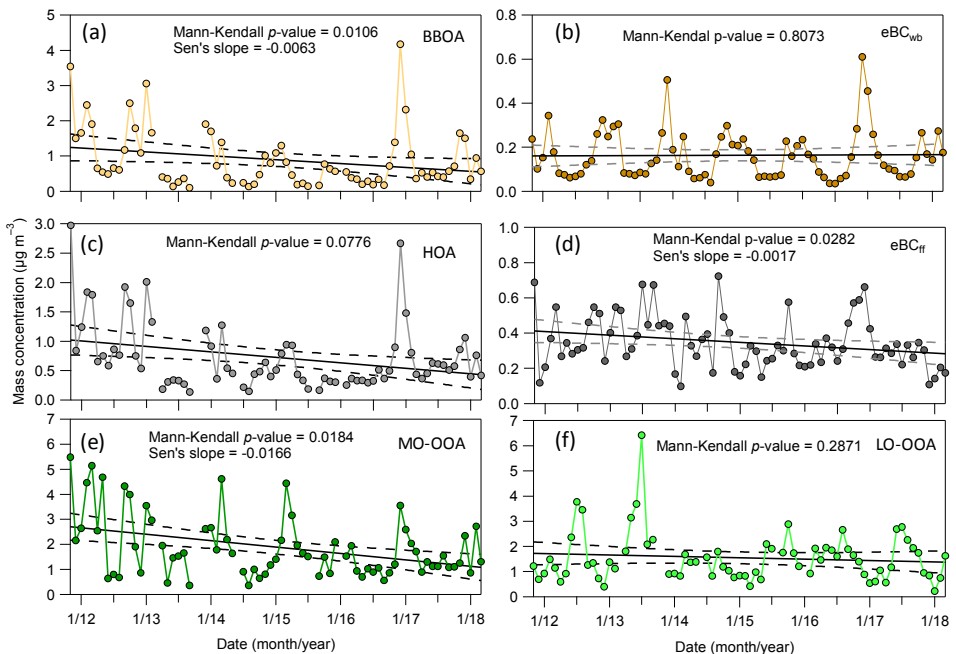


**Figure 6.** 6+-year trends of four OA factors. The (seasonal) Mann-Kendall testes associated

with estimated Sen's slope (only given here when trends are assumed significant, i.e., if MK p-

value < 0.05) were used for the trend analysis.


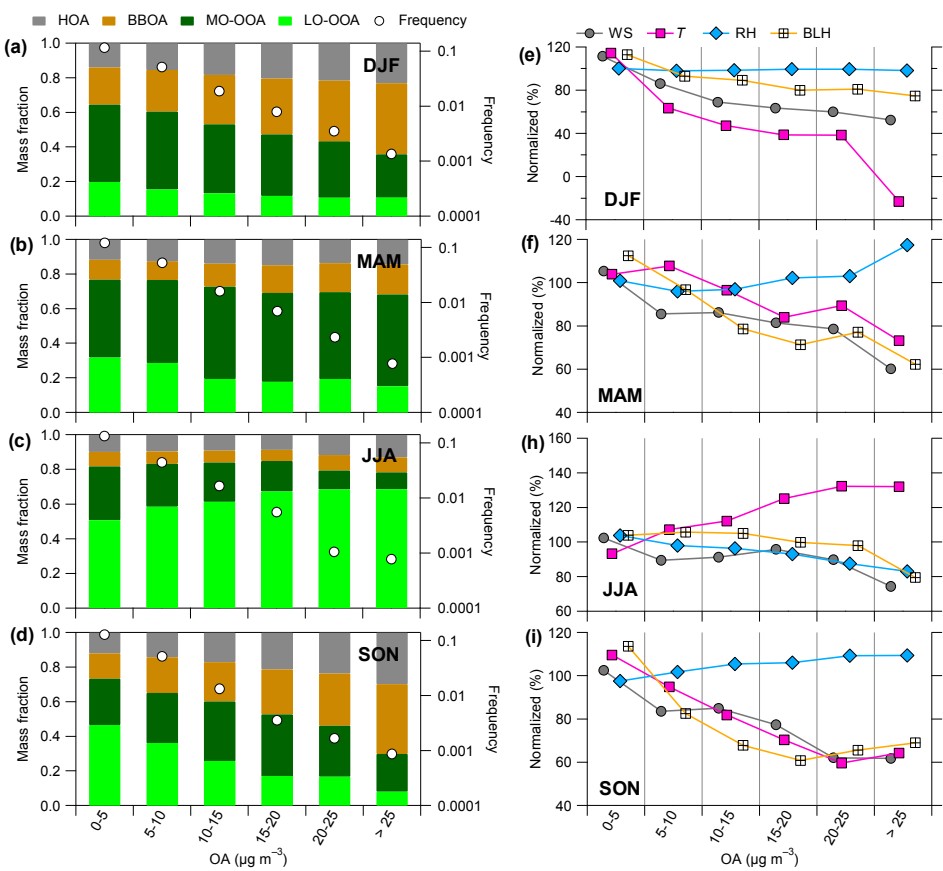


**Figure 7.** (a-d) mass fraction of OA factors and (e-i) meteorological parameters (i.e., WS, *T*, RH,
and BLH) as a function of OA mass loadings in four seasons: winter (DJF), spring (MAM),
summer (JJA), and fall (SON), along with frequency distributions (white circle points). The
percent change of all meteorological parameters was normalized based on the average values
over the 6⁺-years period considered here.



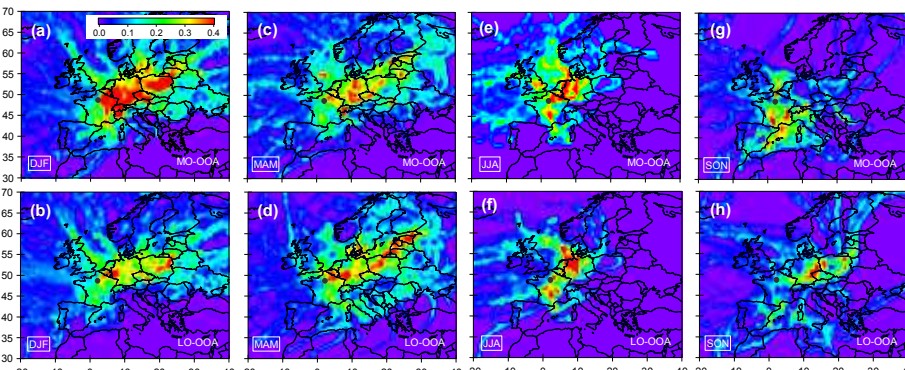


**Figure 8.** Maps for potential source origins of regional transports that may contribute to SOA (including MO-OOA and LO-OOA) burdens at SIRTA. Observed data points with wind speed (less than 4 m s$^{-1}$) and in the presence of precipitation events are filtered for the PSCF calculation. Black solid point in each plot presents the location of the sampling site.