# Peer review of "Six-year source apportionment of submicron organic aerosols from near- continuous measurements at SIRTA (Paris area, France)"

_Atmospheric Chemistry and Physics, 2019_

## Referee Comment (RC1) · Anonymous Referee #1 · 28 Sep 2019

General comment: The manuscript presents the results from 6 years of almost uninterrupted ACSM measurements in the metro area of Paris, and their analysis for OA sources-fractions, inter-annual trends and potential geographic origins. The study is unique as it combines high temporal resolution with a long timeframe, allowing for the examination of variability on different scales. Long-term studies on aerosol characterization and source apportionment are really sparse, so these results are particularly valuable. The paper is generally well written, however the discussion could be expanded with respect to underlying processes of OA formation. The suggested effect of biomass burning on the two resolved OOA factors should be clarified and explained better. The impact of urban emissions and their transport to the receptor site of the

study is minimally presented and should be argued in more detail. Finally, the study presents a real opportunity to assess the actual impact of emission reduction policies enacted in European countries during the last decade. Even though the authors briefly discuss these, I believe that these implications require more focus. Addressing these issues and also the specific comments listed below, the authors can provide a substantially improved version of the manuscript.

Specific comments: Lines 19-20: By now there are numerous annual high-res studies on chemical composition. What is really scarce is long-term studies of the kind. Put emphasis on the long-term aspect here. Line 23: Define here what type of background site (e.g. urban, suburban etc.). Lines 47-56: More or less well known facts. I would suggest shortening and instead adding the more recent advances in understanding the SOA processes. Line 59: Duration of cited studies is up to two years, so better rephrase "multi-year". Also consider citing some filter-based long-term studies for carbonaceous aerosol characterization. Line 61-65: Mention the importance of the high temporal resolution aspect for the study of OA aerosols. Lines 79-82: There is also the SV-LV categorization. However, there is no need to include all the alternative characterizations here. Lines 86-88: Repetitive, can be removed. Line 89-93: Here the transition from emission reductions to the need for long-term observation isn't very clear. Rephrase or omit this, the necessity for long-term has been already stated. Line 93: Probably "robust technology" isn't correct here. Correct this to indicate why the ACSM is more suitable for long-term unattended operation. Line 98: Again, 2 years don't likely qualify for "long-term". Use another term (e.g. "time-extended"). Lines 110-119: Probably the identification and attribution of potential long-terms should be defined as the primary objective. Line 111: Not clear what you mean here. Specify the duration of each PMF analysis and how these results have been integrated for the 6-year period. Lines 122-128: Although references are provided, some details are necessary here regarding siting, especially regarding the traffic and residential characteristics in the surrounding area. Given the distance from the city, this is important for understanding the origins and role of POA. Line 131-133: Provide data capture in

%. Line 145: Add information about corrections for particle collection efficiency. Lines 151-168: Is there any inter-comparison information between AE31-AE33? Discuss the uncertainties expected due to different instrumentation. Lines 166-167: How are these AAE values selected? Based on literature data or some analysis specific to BC aerosols in the area? Lines 184-185: Although it is not the main focus in the study, are there any results from filter comparisons regarding the remaining anions quantified by the ACSM? Sulfate and nitrate concentrations are used in the subsequent analysis. Line 190: Indicate the methodology for the estimation of BLH. Line 195: All the underlying PMF methodological principles are well-known and ubiquitously present in related literature. I suggest removing the details and equations and keep only the specific parametrizations you applied in your analysis. Line 211: Mention why you restricted m/z up to 100 (naphthalene interference etc.). Line 243-247: This sensitivity analysis would be more complete if you had also checked against longer than tri-monthly time windows, to confirm the intuitive approach of seasonal variability. Line 262: Briefly describe the model and discuss temperature related uncertainties. Line 287: The t-test is of minor importance for trend analysis. Also, what is meant by censored data? Better remove the sentence. Lines 290-295: This part has to be clarified. First specify if you use the Theil-Sen estimator for slope estimation. Second, the Kruskal-Wallis test may not suffice to comprehensively assess seasonality for the purposes of trend analysis (it is also not clear which category is used for the test; month, season? any post-hoc evaluations applied? at which level are you testing for significance). It would be better to apply the MK and TS tests to deseasonalized data in all respects. Line 319: Since the choice of weights is empirical and probably study-specific, I suggest removing this. Line 323: The approach here regarding the absence of the COA factor is consistent with past results for ACSM studies in Paris. However, AMS results have indicated that occasionally the COA contribution could be comparable to that of pure BBOA. If it is understood here that COA is incorporated in the BBOA factor, the extent of its potential participation in BBOA should be discussed. Line 367: Correlations with eBCwb and eBCff should be utilized for verification of BBOA, HOA and especially

the BB-related MO-OOA. Check time-lagged cross-correlations between eBCwb and MO-OOA for indications of the aging process. You should also use more your OC, EC data (specifically the OC/EC ratios) for validation of the different OA factors. Lines 377-380: This is somewhat of a stretch. The morning peaks are most probably related to modelling uncertainties. Support with references or remove. However, it might be worth mentioning the highest levels observed during weekends for the BB-related parameters, probably due to recreational use of wood-burning. Lines 381-400: Again, examine the HOA-NOx correlations. Lines 388-390: Is this pattern constant across all seasons? Is there a possibility of HOA emissions from heating oil combustion in winter (if this is a significant source in the Paris region)? Lines 397-400: This is in contrast with the primary nature of hydrocarbon-like OA Consider if this result warrants indicating that extracted HOA in fall/winter is a mixed-factor (in abstract-conclusions). Lines 401-408: Indicate the correlations of MO-OOA with sulfate. Lines 423-427: These have to be reasoned against the fact that maximum LO-OOA levels are observed during nighttime. Line 429: What do you mean by unclear formation schemes? Explain if substantial BSOA formation is plausible based on general vegetation characteristics in the area. Lines 440-443: Rudimentary comments, remove. Lines 453-456: This is a further indication that you should probably deaseasonalize HOA as well. Line 459: Test for significance level. Lines 485-487: This is not sufficient to prove that MO-OOA is BB-related. You should examine the associations with BB-tracers. Lines 487-489: There is number of studies that have associated BBOA with the less oxidized OA fraction as a result of rapid processing during nighttime conditions. The apparent different mechanism here (higher degree of aerosol processing) should be discussed. Lines 490-494: Could low temperatures be associated with increased precursor emissions from biomass burning for heating? Lines 511-514: A major influence from biomass burning has been mentioned in line 417, however it is not considered here. Line 522: I suggest that you perform the trend analysis also for total OA concentrations as well as for the ACSM-derived submicron aerosol concentrations. This can be important from a regulatory standpoint. Also provide numerical results for emission reductions

during the study period, based on national and regional emission inventories. Lines 529-531: Add a reference regarding the relative dependence of OA, BC emissions on woodstove efficiency. Lines 533-535: Check if this exclusion is necessary when using the seasonal test for trend. Lines 544-545: Give the regional character of MO-OOA, you should probably take into account the impact of emission variability on a much larger spatial scale. Lines 562-565: I think that you should formulate this argument the other way round. Lines 610-612: The role of photochemical processing in SOA formation has to be considered here. Line 594: Results from S14 on primary OA should be discussed in more detail. The impact of the city is downplayed, when it should be a primary feature of the study. Line 595: "more stable conditions with anticyclonic conditions". Unclear, clarify. Also add a reference for the synoptic meteorology of the Paris region. Lines 599-601: Based on Fig. 8A, could it be the case that the BB-associations observed during winter for this factor is related to processed BB aerosols originating in central-eastern Europe? Line 610: Figure 8g is essentially the only one presenting a contrasting pattern. Is this association with the Southern trajectories source-related or due to climatology? Lines 610-612: This is very speculative at present. Support with arguments or remove. Line 634: Indicate possible mitigation measures on the local administrative scale. Also that residential biomass burning is assuming Europe-wide importance as a pollution source, but remains largely unregulated. Figure S1a: Check if intercept is statistically significant. If not, run it through the origin. Also not sure that the color scale is informative here. Figure S14: I suggest keeping only the primary factors, move it to the main text and expand the discussion for local sources. Also include wind roses to show the relative prevalence of wind directions.

Technical edits: Line 56: Start new paragraph ("Although..."). Line 130: "...using a quadrupole ACSM. These measurements were performed...". Line 131: Delete "during the investigated period". Lines 133-135: Already mentioned, remove. Line 218: Delete "recently". Line 221: Delete "so-called". Line 231-234: Check citation here. Does this study use PTR-MS? Line 235: "of several". Line 265: "ambient T". Line 387: "ratios are". Line 503: "in more detail" Line 634: "strategies". Line 638:

"contributions to total OA". Line 641: "contributions to OA in wintertime".

---

## Referee Comment (RC2) · Anonymous Referee #2 · 6 Oct 2019

The study represents a multi-year source apportionment of submicron organic aerosol in a regional background site of the Paris metropolital area. 6-year high temporal resolution data from a quadrupole Aerosol Chemical Speciation Monitor (Q-ACSM) are used along with aethalometer data in order to distinguish between different sources contributing to OA loadings during the different seasons. Overall, two primary and two secodnary factors are selected to be representative for the whole measurement period. Primary factors comprise mainly hydrocarbonl-like OA (HOA) and biomass burning OA (BBOA) with both factors exhibiting clear sasonal variability with maxima during winter-fall and minima during summer-spring. Two oxygenated OA factors are also derived, one more- and one less-oxidized (MO-OOA and LO-OOA, respectively). The

MO-OOA also exhibits higher concentrations during wintertime, suggesting common sources from combustion sources and also possible transportation from northeastern Europe, while LO-OOA exhibits higher concentrations and contributions to total OA during summertime, associated with secondary OA formation processes involving biogenic precursors. Finally, multi-annual trend analyses showed a decreasing trend solely for MO-OOA during these 6 years, while very limited or insignificant decreasing trend for the primary OA is observed.

The paper is well written and easy to follow, though there are some issues and more thorough discussion should be made in specific sections. Other than that the paper can be recommended for publication after addressing the issues listed below.

Specific comments:

1) More information about the ACSM measurements and data analysis should be provided: - L145-148: Was there a collection efficiency correction applied?? Was a constant CE used or a chemical composition dependent one e.g. Middlebrook et al. (2012)?

- L184-185: How do the ACSM data compare to the filter measurements? E.g. sulfate, nitrate and ammonium, since they are used further on in the study.

2) PMF analysis: Most of the details in section 3.1 can be omitted, at least the basic principles. On the other hand, more information should be provided for the selection of the specific solutions. E.g. L223-224 what are the final a-values used to constrain POA? In Fig. 1 a=0.21 and a=0.22 for HOA and BBOA are shown, respectively, why are the specific values selected?

3) A more thorough discussion should be made concerning the existence or not of COA. The provided spectra are clearly very different, as obviously the constrained approach is used. When performing a non-constrained run, is there a distinguishable COA factor obtained? Or is it mixed with the BBOA? Furthermore, as COA is con-

sidered to be part of BBOA in this study (if I am not mistaken), and since BBOA concentrations seem really low during summer (Fig. 5), can it be that this BBOA during summer, is indeed the "product" of the source apportionment technique but representing actually COA? Because which primary BB sources can contribute to the site during summertime?

4) More attention should be given to the hypotheses of the origin of the different factors, e.g. L 390-395 HOA considered as a mixture of traffic and biomass burning. Could it be that instead of BB, HOA could be considered as more of a mixture between traffic and combustion from central heating units?

Technical corrections:

L394-395 Rephrase

L488-489 More recent studies also report part of the low-volatility (more oxidized) OOA originating from primary combustion sources (e.g. Stavroulas et al., 2019).

References

1) Middlebrook, A. M., Bahreini, R., Jimenez, J. L., and Canagaratna, M. R.: Evaluation of Composition-Dependent Collection Efficiencies for the Aerodyne Aerosol Mass Spectrometer using Field Data, Aerosol Sci. Technol., 46, 258–271, https://doi.org/10.1080/02786826.2011.620041, 2012.

2) Stavroulas, I., Bougiatioti, A., Grivas, G., Paraskevopoulou, D., Tsagkaraki, M., Zarmpas, P., Liakakou, E., Gerasopoulos, E., and Mihalopoulos, N.: Sources and processes that control the submicron organic aerosol composition in an urban Mediterranean environment (Athens): a high temporal-resolution chemical composition measurement study, Atmos. Chem. Phys., 19, 901–919, https://doi.org/10.5194/acp-19-901-2019, 2019.

---

## Author Comment (AC1) · 28 Oct 2019

Dear Editor,

Thank you very much for sending us the reviewers' constructive comments on our manuscript. We have revised the manuscript accordingly. Listed below is our detailed responses to the reviewers' comments, which are listed in blue. The corresponding modification in the revised manuscript is highlighted *in red and italic text*.

Sincerely yours,

On behalf of the authors,

Yunjiang Zhang and Olivier Favez

**Response to reviewer #1:**

General comment: The manuscript presents the results from 6 years of almost uninterrupted ACSM measurements in the metro area of Paris, and their analysis for OA sources-fractions, inter-annual trends and potential geographic origins. The study is unique as it combines high temporal resolution with a long timeframe, allowing for the examination of variability on different scales. Long-term studies on aerosol characterization and source apportionment are really sparse, so these results are particularly valuable. The paper is generally well written, however the discussion could be expanded with respect to underlying processes of OA formation.

We sincerely thank the reviewer for his or her thoughtful and helpful comments and suggestions.

The suggested effect of biomass burning on the two resolved OOA factors should be clarified and explained better. The impact of urban emissions and their transport to the receptor site of the study is minimally presented and should be argued in more detail.

More discussion about impact of biomass burning on the OOA factors has been added in the revised manuscript with following:

"*... As shown in Figure S15, the correlations between MO-OOA and sulfate are found to be strongly BBOA- and wind speed-dependent. For high wind speed and low BBOA concentrations, the mean MO-OOA-to-sulfate ratio is close to 1, while it reaches up to 8 under high BBOA and low-to-medium wind speed. This is consistent with the assumption of an enhancement of MO-OOA formation in the presence of substantial biomass burning emissions, which have been reported as a major anthropogenic SOA source (Heringa et al., 2011; Tiitta et al., 2016; Bertrand et al., 2017;* Stavroulas et al., 2019; Daellenbach et al., 2019) Actually, both MO-OOA and LO-OOA factors may be significantly influenced by wood burning emissions as they are displaying similar correlations with eBC_{wb} for highest MO-OOA-to-sulfate ratios during wintertime (Figure 6).*"

[Figure]

***Figure 6.*** *Correlations between SOA factors (MO-OOA and LO-OOA) with a BB-related tracer (eBC$_{wb}$) during wintertime. The color-coded solid circle points (in a and b) are the data points corresponding to high ratios of [MO-OOA]-to-[SO$_4$] (more than 8), for which the curve fits are performed.*

Potential impacts of urban emissions (or local scales) and their transport to the sampling site have been further investigated as follows:

*"Figure 9 presents seasonal wind-dependent variations of the POA factors. Wind rose plots (Figure 9a) present the greatest frequency (up to approximate 32%) of winds associated with southwestern wind direction in all seasons. The percentages (about 10-15%) of wind direction from the West-North-East regions are comparable in spring. As presented in Figure 9b, BBOA shows high concentrations associated with the eastern wind sectors, except in summer, which is in agreement with the location of the residential area on the East and West parts of SIRTA. Compared to other seasons, high concentrations of BBOA are also observed linking to western wind sectors in winter, which may imply more intense biomass burning from larger scales during colder months. As discussed above, HOA is a mixed factor with biomass burning aerosols during wintertime, which therefore presents a similar wind-dependent pattern as BBOA (Figure 9c). In spring, summer and fall, HOA presents a distinct pattern with high concentrations associated with northeastern wind sectors from urban area of Paris, suggesting that the short-range transports from the urban Paris area may strongly impact the HOA concentrations at SIRTA.''*

[Figure]

**Figure 9.** *Seasonal wind dependence of POA factors. (a) Wind rose plots color-coded by wind speed (m s⁻¹), and (b) BBOA and (c) HOA, color-coded by mass concentrations (µg m⁻³).*

Finally, the study presents a real opportunity to assess the actual impact of emission reduction policies enacted in European countries during the last decade. Even though the authors briefly discuss these, I believe that these implications require more focus.

Although the investigated dataset is already rather long, it may not be sufficient for the thorough trends' evaluation. However, to address the reviewer comment we extended the discussion on the potential impact of European air-pollution mitigation on different aerosol chemical speciation as follows:

*"The trends are more significant for total OA (p < 0.002, about -382 ng m⁻³ yr⁻¹), as well as for nitrate (p of about 0.01, and approximately -145 ng m⁻³ yr⁻¹) and total PM₁ (p of about 0.002, and approximately -644 ng m⁻³ yr⁻¹), than for sulfate (with p value around 0.5117). In addition to SO₂ emitted from anthropogenic sources (e.g., industrial and shipping emissions)*

*(Hoesly et al., 2018), natural sources (e.g., volcanic emissions) (Boichu et al., 2019) could also influence sulfate budget in western Europe. This suggests that regional aerosol chemistry modeling simulations by using different $SO_2$ emission sectors may help to further explain the temporal trends of sulfate. Meanwhile, it should be noted that the sulfate trend here could be probably influenced by the ACSM measurement uncertainties (Crenn et al., 2015; Freney et al., 2019), which however could not be fully quantified here. Overall, these decreasing trends could reflect the response of the PM concentrations to the decrease in anthropogenic source emissions during these last years in Europe. Reduction in $NO_x$ (-19%) came with a negligible change in $NH_3$ emissions (+2%) over the French region during recent years (2012-2017), which may support that the decreasing trend in particulate nitrate was likely driven by the $NO_x$ emissions control in the Paris region (CITEPA, 2018). A continuous effort to reduce POA emissions and SOA precursors (VOCs) may lead to the decrease to the total both OA and fine PM budgets (EMEP, 2016; CITEPA, 2018)."*

[Figure]

**Figure 7.** *Temporal trends of monthly mass concentrations of different chemical speciation, including $eBC_{wb}$, $eBC_{ff}$, $SO_4$ (sulfate), $NO_3$ (nitrate), four OA factors, total OA, and total $PM_1$ (the sum of $NR$-$PM_1$ and eBC). The (seasonal) Mann-Kendall testes associated with estimated Sen's slope ($\mu g\ m^{-3}$ per year) were used for the trend analysis.*

Moreover, we had also provided some implications for European emission reduction policies or perspectives for the future work that could be done to further understand the relation between emission control strategies and air ambient concentrations of aerosol composition based on the present study. For example (in conclusion section):

*"... BBOA presented a statistically significant seasonal pattern with highest concentrations during cold months, due to residential wood burning emissions. The contribution of BBOA increased with increasing concentration of OA mass in winter and fall – along with decreasing boundary layer height and wind speed – highlighting the importance of biomass burning emissions for OA pollution under stagnant meteorological conditions. HOA presented temporal variations similar to BBOA in cold months, which was partly related to the fact that wood burning emissions also contributed to HOA burden. BBOA and HOA exhibited very limited (< - 100 ng m$^{-3}$ yr$^{-1}$) or not significant trends (at the 5% significance level) during the 6$^{+}$-years investigated period. **These results imply that specific mitigation strategies (e.g., emission control), especially for residential wood burning, are still necessary for substantial improvement of air quality in cold season in the Paris region. Moreover, residential biomass burning emissions could be assumed as an important air-pollution source over western Europe, while such a source remains largely unregulated.***

*LO-OOA and MO-OOA presented different seasonal variations, reflecting different formation mechanisms and/or precursor sources. LO-OOA displayed a pronounced seasonal cycle, with highest contributions to total OA in summer (50-66 %) and lowest ones in winter (12-19 %). Enhanced LO-OOA production during the warm season was assessed to be mainly driven by biogenic SOA formation. This factor showed no significant long-term trend for the studied period. MO-OOA presented higher contribution to OA in wintertime (35-51 %) and springtime (32-62 %) than during the rest of the year. PSCF analyses suggested a high probability of MO-OOA long-range transport from northeastern Europe towards the Paris region. MO-OOA displayed a significant (p<0.05) decreasing trend (of about -175 ng m$^{-3}$ yr$^{-1}$), **which might reflect the effect of emission control strategy of anthropogenic SOA precursors at the regional scale over the last decade. However, future work is needed to fully understand chemical properties of these SOA factors corresponding to different origins over different seasons in the Paris region and to quantify the impact of emission control on ambient SOA burden."***

Addressing these issues and also the specific comments listed below, the authors can provide a substantially improved version of the manuscript.

Thanks again for the reviewer's comments and suggestions. We have revised the manuscript

**Specific comments:**

Lines 19-20: By now there are numerous annual high-res studies on chemical composition. What is really scarce is long-term studies of the kind. Put emphasis on the long-term aspect here.

We agree with the reviewer. This sentence has been revised with:

*"However, highly-time resolved long-term characterizations of their composition and sources in ambient air are still very limited due to challenging continuous observations."*

Line 23: Define here what type of background site (e.g. urban, suburban etc.).

It has been defined by:

*"... at a peri-urban background site of the Paris region (France)."*

Lines 47-56: More or less well known facts. I would suggest shortening and instead adding the more recent advances in understanding the SOA processes.

Those sentences have been revised as follows:

*"Organic aerosol (OA) particles account for a large mass fraction of submicron aerosol ($PM_1$) in the atmosphere (Zhang et al., 2007) and play a key role in regional air pollution and climate (Boucher et al., 2013). Primary OA (POA) originates from direct emissions of primary sources (e.g., fossil-fuel and biomass combustion). Secondary OA (SOA) is formed from atmospheric oxidation processes of gas precursors, i.e., volatile organic compounds (VOCs) (Kroll and Seinfeld, 2008; Hallquist et al., 2009; Nozière et al., 2015). Some typical SOA formation processes in the atmosphere, such as photochemistry (Xu et al., 2017), aqueous-phase oxidation (Gilardoni et al., 2016), and heterogeneous reaction (Xu et al., 2015), are observed. Due to their multiplicity and complexity, these various sources and physicochemical mechanisms remain poorly documented and understood."*

Line 59: Duration of cited studies is up to two years, so better rephrase "multi-year". Also consider citing some filter-based long-term studies for carbonaceous aerosol characterization.

It has been rephrased by "long-term". And some filter-based long-term studies have been cited here.

It now reads as follows:

*"Although numerous time-limited field campaigns allowed to greatly improve our knowledge of OA properties in the last decade (e.g., Jimenez et al., 2009; Lanz et al., 2010; Zhang et al., 2011; Shrivastava et al., 2017; Li et al., 2017; Srivastava et al., 2018a, and references therein), similar studies performed on a* long-term *scale remain scarce and particularly challenging (Fröhlich et al., 2015a; Schlag et al., 2016;* Bozzetti et al., 2017; Daellenbach et al., 2017; *Sun et al., 2018). ..."*

Line 61-65: Mention the importance of the high temporal resolution aspect for the study of OA aerosols.

We have revised the sentence to highlight the importance of the high temporal resolution observation. It now reads as follows:

*"Long-term observations* with high temporal resolution *and source apportionment of OA are nevertheless necessary to better quantify the contribution of airborne OA particles to air quality and to set-up scientifically-sound emission control strategies. They can also contribute to a better understanding of the atmospheric fate of OA and reduce uncertainties associated with its (in)direct radiative forcing."*

In addition, with respect to this comment, the title of the manuscript has been changed as follows:

*"Six-year source apportionment of submicron organic aerosols from near-continuous* highly time-resolved *measurements at SIRTA (Paris area, France)"*

Lines 79-82: There is also the SV-LV categorization. However, there is no need to include all the alternative characterizations here.

We have revised this sentence. It now reads as follows:

*"OOA can be further separated into different fractions, being for instance classified according to its atmospheric ageing described as more oxidized (MO-OOA) or less oxidized (LO-OOA) compared to each other (Jimenez et al., 2009; Ng et al., 2011a; Sun et al., 2018)."*

Lines 86-88: Repetitive, can be removed.

This repetitive information has been removed. It now reads as follows:

*"... Such source apportionment has the potential to assess the efficiency of air pollution mitigation by current emission control strategies."*

Line 89-93: Here the transition from emission reductions to the need for long-term observation isn't very clear. Rephrase or omit this, the necessity for long-term has been already stated.

It has been omitted.

Line 93: Probably "robust technology" isn't correct here. Correct this to indicate why the ACSM is more suitable for long-term unattended operation.

It has been modified such as:

*"Based on better suited for long-term monitoring applications due to lower cost and easier maintenance than AMS, an aerosol chemical speciation monitor (ACSM) has been designed to provide continuous measurements of the main non-refractory chemical species within submicron aerosols (Ng et al., 2011b)."*

Line 98: Again, 2 years don't likely qualify for "long-term". Use another term (e.g. "timeextended").

It has been changed by "time-extended".

*"... So far, several time-extended OA source apportionment studies have been reported based on ACSM measurements at various sites (Canonaco et al., 2015; Fröhlich et al., 2015a; Schlag et al., 2016; Reyes-Villegas et al., 2016; Rattanavaraha et al., 2017; Sun et al., 2018). ..."*

Lines 110-119: Probably the identification and attribution of potential long-terms should be defined as the primary objective.

It has been revised accordingly. It now reads as follows:

*"...In the present study, main OA factors were identified and quantified from seasonal PMF analyses (total 25 seasons) on the $6^+$-year ACSM datasets, with the objective of understanding sources and long-term temporal trends of these factors. ..."*

Line 111: Not clear what you mean here. Specify the duration of each PMF analysis and how these results have been integrated for the 6-year period.

Sorry for confusion. This sentence has been revised as follows:

*"... In the present study, main OA factors were identified and quantified from seasonal PMF analyses (total 25 seasons) on the $6^+$-year ACSM datasets, with the objective of understanding sources and long-term temporal trends of these factors."*

Lines 122-128: Although references are provided, some details are necessary here regarding siting, especially regarding the traffic and residential characteristics in the surrounding area. Given the distance from the city, this is important for understanding the origins and role of POA.

The description proposed about this in the introduction has been extended as follows:

*"The longest ACSM timeseries recorded so far (from end of 2011 onwards) is used here to investigate OA sources at a regional background site of the Paris region (France), which is one*

*of the largest urbanized regions in Europe. It has already been demonstrated that OA plays a dominant role in controlling atmospheric pollution in this region (Bressi et al., 2013; Petit et al., 2015). Furthermore, time-limited (typically, 1–2 months) measurement campaigns demonstrated that primary fine aerosols are mainly influenced there by traffic emissions all over the year and residential wood burning during cold seasons, while secondary aerosols originate from both local production and regional transports (Sciare et al., 2011; Crippa et al., 2013a, Crippa et al., 2013b, Petit et al., 2014; Srivastava et al., 2018b).* *Furthermore, such a background site can be considered as representative of air quality at a regional scale, including neighboring northwestern countries (Bressi et al., 2013; Bressi et al., 2014).**"*

Line 131-133: Provide data capture in %.

The percentage has been given. It now reads:

*"Over the whole investigated period, the data capture was of about 87%, and* *missing data is corresponding to two field campaigns performed elsewhere (in fall 2012 and March 2013) and to few technical breakdown and maintenance periods."*

Line 145: Add information about corrections for particle collection efficiency.

Information about the collection efficiency has been added with:

*"The composition-dependent collection efficiency correction recommended by Middlebrook et al. (2012) has been applied to the whole ACSM data used here."*

Lines 151-168: Is there any inter-comparison information between AE31-AE33? Discuss the uncertainties expected due to different instrumentation.

Yes, there is. The related discussion has been added in the revised manuscript with:

*"An excellent agreement ($r^2$=0.89, slope=1.006±0.006) between AE31 and AE33 for measuring eBC mass concentrations has been demonstrated by Drinovec et al. (2015), suggesting negligible influence of measurement uncertainties between the two mode instruments on quantification of eBC concentrations."*

Lines 166-167: How are these AAE values selected? Based on literature data or some analysis specific to BC aerosols in the area?

The AAE values for different types of eBC, i.e., fossil fuel and wood burning were estimated by the AAE distributions observed in our study (see figure below). Meanwhile, those values are in a good agreement with the recommended values by Zotter et al. (2017).

[Figure]

**Figure R1.** Frequency distribution of ambient AAE values calculated at different paired-wavelengths (i.e., 370 and 950 nm, as well as 470 and 950 nm) and for the 6 wavelengths comprised between 470 and 950 nm.

Lines 184-185: Although it is not the main focus in the study, are there any results from filter comparisons regarding the remaining anions quantified by the ACSM? Sulfate and nitrate concentrations are used in the subsequent analysis.

The ACSM data has been regularly compared to filter-based measurements during field campaigns at SIRTA. An example of obtained results, corresponding to a 2016 springtime campaign (from March 3 to April 22), is now given in the manuscript:

*"The accuracy of these ACSM measurements and the overall good working conditions of the instrument were verified through the participation to the ACTRIS ACSM intercomparison exercises that took place at SIRTA in November - December 2013 (Crenn et al., 2015; Fröhlich*

*et al., 2015b) and March - April 2016 (Freney et al., 2019 and Figure S1)."*

[Figure]

**Figure S1.** *Correlations between the ACSM and 4-h PM$_1$ filter-based measurements for main inorganic species in Spring 2016 (from March 3 to April 22). Satisfactory correlation coefficients (r$^2$ = 0.77 - 0.93) were obtained for each species. For nitrate, a slope of 1.27 is observed, which can be partially explained by the fact that ammonium nitrate is semi-volatile, possibly leading to negative sampling artefacts in off-line measurements (Sciare et al., 2007; Ianniello et al., 2011). Consistently, ammonium also presents a slope higher than 1 (i.e., 1.17), but lower than the slope obtained for nitrate. This is in good agreement with ammonium being mainly combined with nitrate and sulfate, and the non-volatility of ammonium sulfate in ambient conditions. Moreover, the slope obtained for sulfate is much closer to 1 (i.e., 1.03). Overall, those results confirm the validity of the calibration parameters (including response factor and relative ions efficiency values) determined during the 2016 intercomparison exercise (Freney et al., 2019).*

Line 190: Indicate the methodology for the estimation of BLH.

We added this information in the revised manuscript:

*"It should be noted that the BLH data was achieved in combining a diagnostic of the surface stability from high-frequency sonic anemometer measurements and light detection and ranging (LIDAR) attenuated backscatter gradients from aerosols and clouds (Pettie et al., 2015; Dupont et al., 2016)."*

Line 195: All the underlying PMF methodological principles are well-known and ubiquitously present in related literature. I suggest removing the details and equations and keep only the specific parametrizations you applied in your analysis.

Description about the PMF methodology has been simplified in the revised manuscript. It now reads as follows:

*"The OA source apportionment was performed using PMF algorithm (Paatero and Tapper, 1994). Organic concentration and error matrices were exported from the ACSM Local software (v 1.5.11.2). Only m/z ranging from 13 to 100 was applied in the PMF analysis due to larger uncertainties for larger m/z ions and large interferences of naphthalene (m/z 128) signals (Sun et al., 2012). Downweighting of the m/z 44-group ions for the PMF model analysis was performed following procedures implemented in the ACSM Local software and following data treatment strategy proposed by Ulbrich et al. (2009). …"*

Line 211: Mention why you restricted *m/z* up to 100 (naphthalene interference etc.).

The reasons why selecting such *m/z* range have been added in the revised manuscript, which is:

*"Only m/z ranging from 13 to 100 was applied in the PMF analysis due to larger uncertainties for larger m/z ions and large interferences of naphthalene (m/z 128) signals (Sun et al., 2012)."*

Line 243-247: This sensitivity analysis would be more complete if you had also checked against longer than tri-monthly time windows, to confirm the intuitive approach of seasonal variability.

We added a new sensitivity test with a longer PMF window (up to 120 days), in order to evaluate the potential influence of season transition periods on PMF performance. To do so (see Figure R2), we run PMF model with adding 15 days before and after the winter (DJF) days, respectively. As Figure S3 shows, the mass concentrations of the 4 factors are in good agreement with other ones resolved from different window sizes, which supports a reasonable strategy of seasonal PMF analysis performed in this study.

In addition, such a strategy of seasonal PMF analysis used in this work is also consistent with

some previous studies (e.g., Fröhlich et al., 2015; Rattanavaraha et al., 2017; Sun et al., 2018; Stavroulas et al., 2019), because SOA factors are seasonal dependence due to the large differences in meteorological conditions and SOA precursors in different seasons. Those seasonal variabilities could provide large uncertainties for the full-year window PMF analysis.

[Figure]

Figure R2. A scheme description for a new PMF window test (120 days).

[Figure]

*Figure S3.* Comparisons of mass concentrations of four OA factors resolved from different PMF windows runs with setting of 15, 30, 60, 90 and 120 days, respectively.

Line 262: Briefly describe the model and discuss temperature related uncertainties.

Brief description about the model and discussion of temperature related uncertainties have been added in the revised manuscript with follows:

"… To do so, BSOA derived from terpene emissions ($BSOA_t$) was taken as a surrogate for total BSOA and the temperature (T) dependence of the $BSOA_t$ formation process yield during summertime was simulated using a terpene emission model (Goldstein et al., 2009; Schurgers et al., 2009; Leaitch et al., 2011 and references therein). *This model is designed to quantify biogenic emissions over global and regional scales. The emission rate (γ) is estimated by an exponential curve function (Eq. 1), which is describing the relation between terpene γ and leaf T. As we assumed changes in leaf T as same as ambient T, which could then result in part of uncertainties for the model calculation. In addition, this T-dependent model reflects vapour pressure changes caused by T, however, changes in vapour pressure due to changes in the concentrations in the storage pool of terpene are not covered by the model (Schurgers et al., 2009). Therefore, this emission model is useful to simulate the short-term emissions because of T changes (Schurgers et al., 2009).*"

Line 287: The t-test is of minor importance for trend analysis. Also, what is meant by censored data? Better remove the sentence.

Censored data has unknown values beyond a bound on either end of the number line or both, which can exist by design.

As suggested by the reviewer, we have removed this sentence.

Lines 290-295: This part has to be clarified. First specify if you use the Theil-Sen estimator for slope estimation. Second, the Kruskal-Wallis test may not suffice to comprehensively assess seasonality for the purposes of trend analysis (it is also not clear which category is used for the test; month, season? Any post-hoc evaluations applied? at which level are you testing for significance). It would be better to apply the MK and TS tests to deseasonalized data in all respects.

This part has been further updated in the revised manuscript. We have specified that we used the Theil-Sen estimator for slope estimation. We applied the Kruskal-Wallis test for monthly

average datasets at the 5% significance level. To further evaluate the differences of the seasonal MK test and the MK test, we have applied the two methods on all datasets. Please see the modification below:

*"The multi-year trends of monthly mean OA factors and total OA, as well as other chemical components (including $eBC_{wb}$, $eBC_{ff}$, nitrate, sulfate and total $PM_1$) were analyzed using the Mann-Kendall (MK) trend test (Mann, 1945). The trend slope was calculated using Theil-Sen estimator (Sen, 1968). Before performing MK trend test, the normality and seasonality of the OA factors were examined, respectively. The normality of the mass concentrations of the OA factors was examined by the Shapiro-Wilk normality test (Shapiro and Wilk, 1965). As a result of the Shapiro-Wilk normality test, all datasets of the mass concentrations of the four OA factors were not normally distributed cases. The MK test associated with Sen's estimator of slope is insensitive to outliers, while it is not appropriate for the chosen dataset with significant seasonality. The Kruskal-Wallis test (Kruskal and Wallis, 1952) was performed to evaluate the seasonality of monthly average datasets at the 5% significance level. If the seasonality of the data is insignificant, the MK test was used for the trend analysis, while the seasonal MK test was then applied for the data with significant seasonality. In addition, to further compare the differences between the MK test and the seasonal MK test in our trend analysis, both methods have been applied for all data sets (see Table S1). The trend computation was performed here using a R trend package (Pohlert, 2018)."*

**Table S1.** *Trend analysis using the Mann-Kendall test and the seasonal Mann-Kendall test, respectively. The seasonality was exanimated using the Kruskal-Wallis test. Bold text values were selected for the quantification of final trends according to Kruskal-Wallis test at the 5% significance level. Note that the HOA factor without (w/o) December data was also tested.*

| Composition | Kruskal-Wallis test (seasonality) | Mann-Kendall test | | Seasonal Mann-Kendall test | |
|---|---|---|---|---|---|
| | | p-value | Sen's slope | p-value | Sen's slope |
| $PM_1$ | 0.1353 | **0.0123** | **-0.6440** | 0.0056 | -0.7552 |
| $SO_4$ | 0.1210 | **0.5117** | **-0.0237** | 0.4562 | -0.0241 |
| $NO_3$ | 0.0000 | 0.0844 | -0.0961 | **0.0106** | **-0.1447** |
| OA | 0.5594 | **0.0028** | **-0.3823** | 0.0078 | -0.4227 |
| HOA | 0.1001 | **0.0776** | **-0.0512** | 0.0554 | -0.0579 |
| HOA (w/o Dec.) | 0.1622 | **0.0300** | **-0.0594** | 0.02302 | -0.0601 |
| BBOA | 0.0000 | 0.0548 | -0.0674 | **0.0106** | **-0.0647** |
| MO-OOA | 0.0234 | 0.0184 | -0.1813 | **0.0431** | **-0.1754** |
| LO-OOA | 0.0001 | 0.7446 | -0.0159 | **0.2871** | **-0.0383** |
| $eBC_{ff}$ | 0.0101 | 0.0282 | -0.0198 | **0.0040** | **-0.0198** |
| $eBC_{wb}$ | 0.0000 | 0.7482 | -0.0014 | **0.8073** | **-0.0006** |

Line 319: Since the choice of weights is empirical and probably study-specific, I suggest removing this.

It has been removed from the revised manuscript.

Line 323: The approach here regarding the absence of the COA factor is consistent with past results for ACSM studies in Paris. However, AMS results have indicated that occasionally the COA contribution could be comparable to that of pure BBOA. If it is understood here that COA is incorporated in the BBOA factor, the extent of its potential participation in BBOA should be discussed.

Although the COA or COA-like factor has been identified in the Paris region from AMS measurements, it is still a great challenge to identify and quantify such cooking factor based on Q-ACSM dataset at SIRTA (a regional background site of the Paris area). It could be due to two possible reasons. First, the contribution of cooking to ambient organic aerosol at SIRTA area was very low or very limited. This is in fact consistent with negligible cooking emission at

SIRTA nearby. On the other hand, such low contribution can substantially lead to large uncertainties on both, Q-ACSM measurements (lower detection resolution than AMS) and subsequent PMF analysis (model uncertainty).

We have added such discussion in the revised manuscript with follows:

"*Figures 1 and S5 present results obtained for the 4- and 5-factor solutions, respectively, for the winter 2017-2018 period, taken here as an example. In both cases, mass spectra were in good agreement with those reported in the literature. However, the COA and BBOA factors are displaying very similar diel patterns, leading to surprisingly good correlations between these two factors (Figure S6). In order to further evaluate possible COA contribution at SIRTA, we applied a m/z-tracer algorithm (Mohr et al., 2012) trying to identify pure cooking aerosol signals (see Figure S7). The distribution of the estimated COA signals is centered about 0, as indicated by the result of Figure S7. This could be probably explained by very little pure cooking influence that could not be quantified by the lower resolution quadrupole ACSM than AMS, which is logically in agreement with negligible cooking source at the sampling site area nearby. Altogether, it could then be concluded that the constrained COA-like aerosols at SIRTA were primarily linked with wood burning emissions, while pure cooking aerosols were probably present in too low loadings to be properly quantified within the present study. This assumption is consistent with conclusions drawn by other studies performed at SIRTA, e.g., based on an online (ACSM) dataset (Petit et al., 2014) and a combining PMF method using online (ACSM) and offline (4-h filter sampling) datasets (Srivastava et al., 2019), as well as other studies showing that the COA factor could not be solely attributed to cooking aerosols (e.g., Freutel et al., 2013, Dall'Osto et al., 2015).*"

[Figure]

**Figure S7.** Distribution of the estimated COA signals by using a m/z-tracer method (Mohr et al., 2012) during the entire period. Briefly, the COA signals can be calculated based on the time series of the signals of *m/z* 55, *m/z* 57, and *m/z* 44 measured by the ACSM (using the equation: $COA_{est} = \left[\frac{\frac{1}{a} \cdot m/z55 - m/z57 + \left(c - \frac{b}{a}\right) \cdot m/z44}{\frac{1}{a} - \frac{1}{d}}\right]$). The parameters in this equation, a, b, c and d, are determined by the corresponding signals intensity in mass spectra of OA factors, which have been clearly described by Mohr et al. (2012).

Line 367: Correlations with eBC$_{wb}$ and eBC$_{ff}$ should be utilized for verification of BBOA, HOA and especially the BB-related MO-OOA.

The two types of eBC fractions have been further used for comparisons with OA factors. The corresponding discussions have been also given in the revised manuscript, as described hereabove (new figure 6) and below:

*"…The correlations of OA factors with their tracers were examined to globally evaluate the 4-factor PMF solution (see Figures S10 and S11). As shown in Figure S10a, HOA is correlated well ($r^2$=0.54) with NOx, a common tracer of primary combustion sources (e.g., traffic emissions). While HOA shows a relatively weaker correlation ($r^2$=0.33) with eBC$_{ff}$ (Figure S10b), this could be explained by two possible reasons, i) uncertainties of Aethalometer model which however could not be evaluated by the present study, and ii) the HOA factor here could not be reprehensive for pure fossil-fuel combustion POA. BBOA presents an overall good correlation ($r^2$=0.50) with eBC$_{wb}$ (Figure S10c), suggesting important influence of wood burning emissions on this factor production. Based on the filter-based dataset, primary OC (POC) and secondary OC (SOC) were calculated using a method of OC-to-EC mass ratio (see Figure S11). Overall, POA (sum of HOA and BBOA) versus POC ($r^2$=0.47) and SOA (sum of*

*LO-OOA and MO-OOA) versus SOC (r²=0.38) have acceptable correlations during the entire filter measurement period. Thus, all of these comparison results could additionally support our "best estimation" for selecting such 4-factor PMF solution across the entire period."*

Check time-lagged cross-correlations between eBC$_{wb}$ and MO-OOA for indications of the aging process.

The cross-correlations between eBC$_{wb}$ and MO-OOA have been checked (see Figure R3). There is a strong correlation at a time delay of about 0, indicating limited time delay between the two variables, i.e., eBC$_{wb}$ and MO-OOA. This suggests the expected aging process of biomass burning SOA could be not measured using such method. Nevertheless, the reviewer suggested a good point that is interesting to be further investigated by other measures, such as aerosol chemical transport models with observational constraints in the future.

[Figure]

**Figure R3.** The cross-correlation series for the two variables, i.e., eBC$_{wb}$ and MO-OOA, in winter during the entire study period.

You should also use more your OC, EC data (specifically the OC/EC ratios) for validation of the different OA factors.

We have applied the OC/EC mass ratio method to apportion primary OC and secondary OC, respectively, to examine PMF OA factors. The related modification has been made in the revised manuscript:

*"... Based on the filter-based dataset, primary OC (POC) and secondary OC (SOC) were*

*calculated using a method of OC-to-EC mass ratio (see Figure S11). Overall, POA (sum of HOA and BBOA) versus POC ($r^2$=0.47) and SOA (sum of LO-OOA and MO-OOA) versus SOC ($r^2$=0.38) have acceptable correlations during the entire filter measurement period. ...''*

[Figure]

***Figure S11.** Intercomparison between POA (resp. SOA) and POC (resp. SOC), where POC and SOC were calculated using the OC/EC mass ratio method. The seasonal minimum OC/EC mass ratios – DJF (2.45), MAM (1.32), JJA (1.52), and SON (1.46) – were assumed being from primary combustion source emissions. Such OC/EC ratio method to has been widely applied to isolate POC and SOC from total OC by numerous previous studies (e.g., Srivastava et al., 2018, and references therein).*

Lines 377-380: This is somewhat of a stretch. The morning peaks are most probably related to modelling uncertainties. Support with references or remove.

The morning peaks related discussion has been removed.

However, it might be worth mentioning the highest levels observed during weekends for the BB-related parameters, probably due to recreational use of wood-burning.

This sentence has been revised. It now reads as follows:

*"... Interestingly, the higher concentrations of eBC$_{wb}$ and BBOA were observed on Saturday and Sunday, reflecting the week-end effect likely due to enhanced residential wood burning emissions."*

Lines 381-400: Again, examine the HOA-NOx correlations.

The correlation between HOA and NOx has been given in the revised manuscript. It can be referred by the comment of "Line 367" above.

Lines 388-390: Is this pattern constant across all seasons? Is there a possibility of HOA emissions from heating oil combustion in winter (if this is a significant source in the Paris region)?

This pattern is not constant across all seasons. Such impact on HOA factor is mainly seen in winter, suggesting influence of residential heating. We cannot evidence how much heating oil emissions might play a substantial role in this (or not). We then added this hypothesis in the revised manuscript, as follows:

"...However, HOA evening peaks occurs globally later than $eBC_{ff}$ and $NO_x$ ones (9:00-10:00 PM vs. 7:00 PM, respectively) and much lower ratios are observed between HOA and $eBC_{ff}$ in the morning than in the evening. This might be partly explained by i) higher eBC traffic emission factor in the morning and/or ii) impacts of residential heating sources, e.g., wood and/or heating oil burning (Lin et al., 2018), on the HOA concentrations in the late evening."

Lines 397-400: This is in contrast with the primary nature of hydrocarbon-like OA. Consider if this result warrants indicating that extracted HOA in fall/winter is a mixed-factor (in abstract-conclusions).

Thanks for the reviewer's comments. We have removed this sentence from the revised manuscript.

Lines 401-408: Indicate the correlations of MO-OOA with sulfate.

The correlation between MO-OOA and sulfate has been indicated in the revised manuscript. It reads as follows:

"... Overall, MO-OOA had a weak correlation ($r^2$=0.23) with sulfate during the entire period,

*supporting their different source origins to some extent. …"*

Lines 423-427: These have to be reasoned against the fact that maximum LO-OOA levels are observed during nighttime.

The diel cycles of the LO-OOA factor, characterized by high concentrations during nighttime, have been further discussed in the revised manuscript following:

*"The diel variations of LO-OOA display higher concentrations during nighttime than daytime (Figure 2d), with relative variations much more pronounced than for the MO-OOA diel pattern, highlighting important roles of nighttime chemistry and/or gas-particle partitioning in the LO-OOA formation. These results support different formation pathways of the two OOA fractions. In addition, LO-OOA and nitrate present different diel cycles, suggestive of different formation processes and sources between each other. Different diel cycles of LO-OOA in different seasons have been also observed, which are further discussed in section 4.2.1."*

In addition, high concentrations of the LO-OOA factor were observed during nighttime and some hours during daytime in summer, which has been also further discussed in the revised manuscript. It reads as follows:

*"… Moreover, high concentrations of the summertime LO-OOA are observed during the two distinct time periods in one day, i.e., early afternoon (around 12:00 – 15:00) and nighttime (around 22:00-05:00), which is different from the diel variations in other seasons with high concentrations only during nighttime (Figure S14). These LO-OOA diel variations may reflect different formation pathways across one day in summer. Photochemical process might dominate the LO-OOA production at daytime, while nighttime chemistry and/or gas-particle partitioning might promote its formation at low T conditions at night. …"*

Line 429: What do you mean by unclear formation schemes? Explain if substantial BSOA formation is plausible based on general vegetation characteristics in the area.

Sorry for confusion. It should be better "unclear formation mechanism".

The presence of the BSOA formation during summertime in the Paris region can be logically supported the seasonal characterization of biogenic contribution (increased up to 30% in summer) to total VOCs loadings observed in Paris area (Baudic et al., 2016), as well as modeling simulations (48 % of BSOA in total SOA) during the MEGAPOLI summer campaign period (Beekmann et al. 2015). Such discussion had been given in the manuscript as follows:

*"Conversely, in summer, this factor may be significantly influenced by BSOA formation (Canonaco et al., 2015; Daellenbach et al., 2017). To investigate this possible origin, we checked if the summertime LO-OOA concentrations at higher daily T were following temperature dependence similar to the one expected for the formation of terpene SOA, as explained in section 3.2. Results of these calculations are presented in Figure 3. The LO-OOA concentrations substantially increase with T, showing a good agreement with the estimated $BSOA_t$ formation exponential profiles. However, when comparing with estimation derived from Eq. (4) (referred to Figure 4), the observed LO-OOA displays substantially higher loadings than estimated $BSOA_t$ at highest concentration range. This could be partly due to the influence of regional transports and atmospheric dilution on aerosol loadings and some possible uncertainties (such as unclear formation mechanism of biogenic SOA at SIRTA), which were not considered in the $BSOA_t$ estimation. These comparison results between observation and estimation indicates that the LO-OOA factor observed in summer might be mainly associated with biogenic sources. **This is aligned with the VOCs seasonal patterns observed in the Paris region (Baudic et al., 2016), although the underlying SOA formation mechanism is still unclear and needs to be further investigated (Beekmann et al. 2015).** Further discussion about seasonality of the LO-OOA factor is given in section 4.2.1."*

Lines 440-443: Rudimentary comments, remove.

Removed.

Lines 453-456: This is a further indication that you should probably deseasonalize HOA as well.

If we are not mistaken, the reviewer's suggestion here is linking to the Mann-Kendall trend test (but not seasonal Mann-Kendall trend test) that should be performed in our study. In fact,

we performed the Mann-Kendall trend test for the HOA factor, since HOA has no significant seasonality (Kruskal-Wallis *p*>0.05). Nevertheless, we have performed both, the Mann-Kendal test and the seasonal Mann-Kendall test, in our trend analysis for all examined species. We have also revised the manuscript accordingly:

*"... The Kruskal-Wallis test (Kruskal and Wallis, 1952) was performed to evaluate the seasonality of monthly average datasets at the 5% significance level. If the seasonality of the data is insignificant, the MK test was used for the trend analysis, while the seasonal MK test was then applied for the data with significant seasonality. In addition, to further compare the differences between the MK test and the seasonal MK test in our trend analysis, both methods have been applied for all data sets (see Table S1). ..."*

Line 459: Test for significance level.

Thanks for the comment. About 1.3-1.5 times higher was observed. The related sentence has been revised by:

*"The evening HOA peak is about 1.3-1.5 times higher than the morning peak in winter and fall seasons when high loadings of BBOA are observed as well."*

Lines 485-487: This is not sufficient to prove that MO-OOA is BB-related. You should examine the associations with BB-tracers. + Lines 511-514: A major influence from biomass burning has been mentioned in line 417, however it is not considered here.

Yes, it has been done as detailed hereabove. Thanks again for helping to clarify this.

Lines 490-494: Could low temperatures be associated with increased precursor emissions from biomass burning for heating?

It could be possible. We have added a new discussion about relation between the potential increased biomass burning emission and enhanced MO-OOA formation at low temperature conditions during wintertime. It reads as follows:

*"As shown in Figure S16, high concentrations of MO-OOA are generally observed at high RH*

*(> 80 %) and low T (< 0 °C) conditions during wintertime. This low air temperature condition could be associated with a possible scenario for increase of the MO-OOA precursors emissions from biomass burning by residential heating activities during wintertime.''*

Line 522: I suggest that you perform the trend analysis also for total OA concentrations as well as for the ACSM-derived submicron aerosol concentrations. This can be important from a regulatory standpoint. Also provide numerical results for emission reductions during the study period, based on national and regional emission inventories.

As suggested, we have performed the trend analysis for OA factors and some others measured by the ACSM. The numerical results obtained from an emission inventory (CITEPA) over French region has been added for the related discussion in the revised manuscript.

Lines 529-531: Add a reference regarding the relative dependence of OA, BC emissions on woodstove efficiency.

The related reference (Saleh et al., 2014) has been added in the revised manuscript.

*Saleh, R., Robinson, E. S., Tkacik, D. S., Ahern, A. T., Liu, S., Aiken, A. C., Sullivan, R. C., Presto, A. A., Dubey, M. K., Yokelson, R. J., Donahue, N. M., and Robinson, A. L.: Brownness of organics in aerosols from biomass burning linked to their black carbon content, Nature Geoscience, 7, 647, 10.1038/ngeo2220, 2014.*

Lines 533-535: Check if this exclusion is necessary when using the seasonal test for trend.

We have checked the seasonality of the HOA factor without the December data, in which a Kruskal-Wallis's p value (0.1622) was observed. Meanwhile, we also performed the test with the seasonal MK trend test, the results from are comparable with ones obtained from the MK trend test (See Table S1).

Lines 544-545: Give the regional character of MO-OOA, you should probably take into account the impact of emission variability on a much larger spatial scale.

It has been revised, which now reads as follows:

*"…MO-OOA shows a significant decreasing trend (p < 0.05) with a Sen's slope of -175 ng m$^{-3}$ per year. Considering the overwhelming secondary origin of this factor, this significant decreasing trend may be partially explained by an overall reduction of anthropogenic VOCs emissions* (-13%) *in France (CITEPA, 2018) and even in a larger spatial scale, e.g., the western European regions, during 2012-2017."*

Lines 562-565: I think that you should formulate this argument the other way round.

This sentence has been revised as follows:

*"… POA contributions gradually increase from 35 % (resp. 27 %) up to 64 % (resp. 70 %) as a function of OA mass concentrations in winter (resp. fall). …"*

Lines 610-612: The role of photochemical processing in SOA formation has to be considered here.

This sentence has been removed according to the comment below on this sentence.

Line 594: Results from S14 on primary OA should be discussed in more detail. The impact of the city is downplayed, when it should be a primary feature of the study.

This part is now further discussed:

*"Figure 9 presents seasonal wind-dependent variations of the POA factors. Wind rose plots (Figure 9a) present the greatest frequency (up to approximate 32%) of winds associated with southwestern wind direction in all seasons. The percentages (about 10-15%) of wind direction from the West-North-East regions are comparable in spring. As presented in Figure 9b, BBOA shows high concentrations associated with the eastern wind sectors, except in summer, which is in agreement with the location of the residential area on the East and West parts of SIRTA. Compared to other seasons, high concentrations of BBOA are also observed linking to western wind sectors in winter, which may imply more intense biomass burning from larger scales during colder months. As discussed above, HOA is a mixed factor with biomass*

*burning aerosols during wintertime, which therefore presents a similar wind-dependent pattern as BBOA (Figure 9c). In spring, summer and fall, HOA presents a distinct pattern with high concentrations associated with northeastern wind sectors from urban area of Paris, suggesting that the short-range transports from the urban Paris area may strongly impact the HOA concentrations at SIRTA.''*

Line 595: "more stable conditions with anticyclonic conditions". Unclear, clarify. Also add a reference for the synoptic meteorology of the Paris region.

Sorry for confusion. We removed this part of the sentence. It now reads as follows:

*"Those results may suggest more intense SOA production and aging processes at regional scale for continental air masses.".*

Lines 599-601: Based on Fig. 8A, could it be the case that the BB-associations observed during winter for this factor is related to processed BB aerosols originating in central-eastern Europe?

This is an interesting point. It could be another possible explanation for the enhanced aged SOA production observed at SIRTA, in addition to influence of regional fossil-fuel combustion sources (e.g., industrial sector). However, we don't have direct evidence to further support such statement. Thus, we rephased this sentence to enlarge the possibilities. It now reads as follows:

"Moreover, the impact of transport from northeastern regions – hosting intense anthropogenic activities (e.g., industries) - onto MO-OOA concentrations may also support a significant anthropogenic origin for this aged SOA factor."

Line 610: Figure 8g is essentially the only one presenting a contrasting pattern. Is this association with the Southern trajectories source-related or due to climatology?

Yes, this can be associated with the back-trajectories of air masses from the southern region. We unfortunately do not have evidence to prove the influence of climatology on that.

Lines 610-612: This is very speculative at present. Support with arguments or remove.

Removed.

Line 634: Indicate possible mitigation measures on the local administrative scale. Also that residential biomass burning is assuming Europe-wide importance as a pollution source, but remains largely unregulated.

Thanks for the reviewers' comment. This has been revised accordingly:

"… *These results imply that specific mitigation* strategies (e.g., emission control), especially for *residential wood burning, are still necessary for substantial improvement of air quality in cold season in the Paris region.* Moreover, residential biomass burning emissions could be assumed as an important air-pollution source over western Europe, while such a source remains largely unregulated."

Figure S1a: Check if intercept is statistically significant. If not, run it through the origin. Also not sure that the color scale is informative here.

We did a test for the significance of the intercept based on linear regression analysis (see Table R1 below). As indicated by the result, the intercept is statistically significant ($p < 0.0001$). However, we did a mistake for the intercept, the value of which should be 0.05 instead of 0.005. We have modified it in the revised manuscript. The color scale can provide useful information to visualize the time-dependent distributions of those data points in the plots.

Table R1. Linear regression analysis on eBC versus EC.

|  | Coefficients | Standard Error | t Stat | P-value |
|---|---|---|---|---|
| Intercept | 0.05 | 0.01 | 6.12 | <0.0001 |
| Slope | 0.94 | 0.01 | 73.29 | <0.0001 |

Figure S14: I suggest keeping only the primary factors, move it to the main text and expand

the discussion for local sources. Also include wind roses to show the relative prevalence of wind directions.

We have removed plots regarding SOA factors. And we also moved the wind dependence of POA factors with expanded discussion in the main text. Wind rose plots have been added in this figure as well (Please see above).

**Technical edits:**

Line 56: Start new paragraph ("Although: : :).

A new paragraph has been made.

Line 130: ": : :using a quadrupole ACSM. These measurements were performed: : :".

Revised.

Line 131: Delete "during the investigated period".

Deleted.

Lines 133-135: Already mentioned, remove.

Removed.

Line 218: Delete "recently".

Removed.

Line 221: Delete "so-called".

Deleted.

Line 231-234: Check citation here. Does this study use PTR-MS?

Thanks for the comment. This not a correct reference (Crippa et al., 2013c) here, while it should be Crippa et al. (2013a).

Crippa, M., Canonaco, F., Slowik, J. G., El Haddad, I., DeCarlo, P. F., Mohr, C., Heringa, M. F., Chirico, R., Marchand, N., Temime-Roussel, B., Abidi, E., Poulain, L., Wiedensohler, A., Baltensperger, U., and Prévôt, A. S. H.: Primary and secondary organic aerosol origin by combined gas-particle phase source apportionment, Atmos. Chem. Phys., 13, 8411-

8426, 10.5194/acp-13-8411-2013, 2013a.

Line 235: "of several".

Revised.

Line 265: "ambient T".

Revised.

Line 387: "ratios are".

Revised.

Line 503: "in more detail"

Revised.

Line 634: "strategies".

Revised.

Line 638: "contributions to total OA".

Revised.

Line 641: "contributions to OA in wintertime".

Revised.

**Response to reviewer #2:**

The study represents a multi-year source apportionment of submicron organic aerosol in a regional background site of the Paris metropolital area. 6-year high temporal resolution data from a quadrupole Aerosol Chemical Speciation Monitor (Q-ACSM) are used along with aethalometer data in order to distinguish between different sources contributing to OA loadings during the different seasons. Overall, two primary and two secondary factors are selected to be representative for the whole measurement period. Primary factors comprise mainly hydrocarbon-like OA (HOA) and biomass burning OA (BBOA) with both factors exhibiting clear seasonal variability with maxima during winter-fall and minima during summer-spring. Two oxygenated OA factors are also derived, one more- and one less-oxidized (MO-OOA and LO-OOA, respectively). The MO-OOA also exhibits higher concentrations during wintertime, suggesting common sources from combustion sources and also possible transportation from northeastern Europe, while LO-OOA exhibits higher concentrations and contributions to total OA during summertime, associated with secondary OA formation processes involving biogenic precursors. Finally, multi-annual trend analyses showed a decreasing trend solely for MO-OOA during these 6 years, while very limited or insignificant decreasing trend for the primary OA is observed.

The paper is well written and easy to follow, though there are some issues and more thorough discussion should be made in specific sections. Other than that the paper can be recommended for publication after addressing the issues listed below.

We thank the reviewer very much for his or her positive and constructive comments. We have revised the manuscript accordingly.

Specific comments:

1) More information about the ACSM measurements and data analysis should be provided: - L145-148: Was there a collection efficiency correction applied?? Was a constant CE used or a chemical composition dependent one e.g. Middlebrook et al. (2012)?

Yes, we applied the composition-dependent CE correction (Middlebrook et al., 2012) for the whole dataset. We have added the sentence below in the revised manuscript:

*"The composition-dependent collection efficiency correction recommended by Middlebrook et al. (2012) has been applied to the whole ACSM data used here."*

- L184-185: How do the ACSM data compare to the filter measurements? E.g. sulfate, nitrate and ammonium, since they are used further on in the study.

We don't have such near-continuous data for inorganic species. However, ACSM data has been regularly compared to filter-based measurements during field campaigns at SIRTA. An example of obtained results, corresponding to a 2016 springtime campaign (from March 3 to April 22), is now given in the manuscript:

*"The accuracy of these ACSM measurements and the overall good working conditions of the instrument were verified through the participation to the ACTRIS ACSM intercomparison exercises that took place at SIRTA in November - December 2013 (Crenn et al., 2015; Fröhlich et al., 2015b) and March - April 2016 (Freney et al., 2019 and Figure S1)."*

[Figure]

***Figure S1.*** *Correlations between the ACSM and 4-h PM$_1$ filter-based measurements for main inorganic species in Spring 2016 (from March 3 to April 22). Satisfactory correlation coefficients (r$^2$ = 0.77 - 0.93) were obtained for each species. For nitrate, a slope of 1.27 was observed, which can be partially explained by the fact that ammonium nitrate is semi-volatile, possibly leading to negative sampling artefacts in off-line measurements (Sciare et al., 2007; Ianniello et al., 2011). Consistently, ammonium also presented a slope higher than 1 (i.e., 1.17), but lower than the slope obtained for nitrate. This is in good agreement with ammonium being mainly combined with nitrate and sulfate, and the non-volatility of ammonium sulfate in ambient conditions. Moreover, the slope obtained for sulfate is much closer to 1 (i.e., 1.03). Overall, those results confirmed the validity of the calibration parameters (including response*

*factor and relative ions efficiency values) determined during the 2016 intercomparison exercise (Freney et al., 2019).*

2) PMF analysis: Most of the details in section 3.1 can be omitted, at least the basic principles. On the other hand, more information should be provided for the selection of the specific solutions. E.g. L223-224 what are the final a-values used to constrain POA? In Fig. 1 a=0.21 and a=0.22 for HOA and BBOA are shown, respectively, why are the specific values selected?

Some details about the PMF model description have been deleted from the revised manuscript.

The final factor solution was achieved from the mean values of 50 repeat runs (with *a* value ranging from 0 to 0.4 to constrain POA factors, i.e., HOA and BBOA) in each PMF analysis for each season (total 25 seasons). Therefore, the *a-value* for HOA (0.21) and BBOA (0.22) in Fig. 1 are the mean values from these 50 repeat runs.

3) A more thorough discussion should be made concerning the existence or not of COA. The provided spectra are clearly very different, as obviously the constrained approach is used. When performing a non-constrained run, is there a distinguishable COA factor obtained?

The more discussion has been made in the revised manuscript. We have revised the manuscript accordingly:

*"Figures 1 and S5 present results obtained for the 4- and 5-factor solutions, respectively, for the winter 2017-2018 period, taken here as an example. In both cases, mass spectra were in good agreement with those reported in the literature. However, the COA and BBOA factors are displaying very similar diel patterns, leading to surprisingly good correlations between these two factors (Figure S6). In order to further evaluate possible COA contribution at SIRTA, we applied a m/z-tracer algorithm (Mohr et al., 2012) trying to identify pure cooking aerosol signals (see Figure S7). The distribution of the estimated COA signals is centered about 0, as indicated by the result of Figure S7. This could be probably explained by very little pure cooking influence that could not be quantified by the lower resolution quadrupole ACSM than AMS, which is logically in agreement with negligible cooking source at the sampling site area nearby. Altogether, it could then be concluded that the constrained COA-like aerosols at SIRTA were*

*primarily linked with wood burning emissions, while pure cooking aerosols were probably present in too low loadings to be properly quantified within the present study. This assumption is consistent with conclusions drawn by other studies performed at SIRTA, e.g., based on an online (ACSM) dataset (Petit et al., 2014) and a combining PMF method using online (ACSM) and offline (4-h filter sampling) datasets (Srivastava et al., 2019), as well as other studies showing that the COA factor could not be solely attributed to cooking aerosols (e.g., Freutel et al., 2013, Dall'Osto et al., 2015)."*

[Figure]

**Figure S7.** Distribution of the estimated COA signals by using a m/z-tracer method (Mohr et al., 2012) during the entire period. Briefly, the COA signals can be calculated based on the time series of the signals of *m/z* 55, *m/z* 57, and *m/z* 44 measured by the ACSM (using the equation: $COA_{est} = \left[\dfrac{\frac{1}{a} \cdot m/z55 - m/z57 + \left(c - \frac{b}{a}\right) \cdot m/z44}{\frac{1}{a} - \frac{1}{d}}\right]$). The parameters in this equation, a, b, c and d, are determined by the corresponding signals intensity in mass spectra of OA factors, which have been clearly described by Mohr et al. (2012).

We also already have performed some tests with unconstrained PMF analysis using a PMF Evaluation Tool (Ulbrich et al., 2009). As shown in Figures R4-6 (as an example), however, there were no COA factor that could be identified by this method. Moreover, we also did another PMF analysis by combining offline (including EC/OC, anions/cations, methanesulfonic acid, oxalate, cellulose combustion markers (levoglucosan, mannosan, and galactosan), 3 polyols (arabitol, sorbitol, and mannitol), 9 polycyclic aromatic hydrocarbons (PAHs), 14 oxy-PAHs, 8 nitro-PAHs and 13 SOA markers and online (ACSM) datasets during 2015 springtime intensive campaign (*Srivastava et al., 2019*). However, cooking source was not identified neither.

Overall, all of those data analysis strategies – i.e., constrained ME-2, unconstrained PMF, and combining PMF with both offline and online datasets, as well as a *m/z*-tracer method, could not substantially identify the COA factor. All those results may support very little or negligible pure cooking organics based on our experimental methodologies, which is consistent well with the fact that very limited cooking source distributes at the sampling site nearby.

[Figure]

Figure R4. PMF analysis of OA mass spectra using a PMF Evaluation Tool (Ulbrich et al., 2009). Left panel plots show mass spectrum of the 4 OA factors, and right panel plots show the corresponding diel variations.

[Figure]

Figure R5. Same as Figure R4, but for 5-factor solution.

[Figure]

Figure R6. Same as Figure R4, but for 6-factor solution.

Or is it mixed with the BBOA? Furthermore, as COA is considered to be part of BBOA in this study (if I am not mistaken), and since BBOA concentrations seem really low during summer (Fig. 5), can it be that this BBOA during summer, is indeed the "product" of the source apportionment technique but representing actually COA? Because which primary BB sources can contribute to the site during summertime?

No, the COA-like factor, tested by the present work, was proved to be mainly mixed with the HOA factor (in the 4-factor solution).

The summertime biomass burning sources could be explained by two potential reasons. First, BBOA might origin from irregular biomass burning emissions, from residential sectors (van Marle et al., 2017) in summer. This could be also supported by some biomass-burning signals indicated by the higher $f_{60}$ values (> 0.3±0.16%) (see Figure R7). In addition to this reason, the low contribution (about 5%) of BBOA to the total OA is comparable to the uncertainty of PMF

model (Ulbrich et al., 2009). In order to better establish the data temporal continuity for all OA factors and to avoid seasonal gap for the BBOA factor in the time series analysis, we therefore kept the summertime BBOA in our study.

[Figure]

**Figure R7.** Relationship between $f_{44}$ and $f_{60}$ (the fraction of m/z44 and m/z60 in total OA signal, respectively) observed in all summers. The read dash line indicates a background level ($f_{60}$ < 0.3 ± 0.16%) for little or negligible BB influence (Cubison et al., 2011; Zhang et al., 2015).

4) More attention should be given to the hypotheses of the origin of the different factors, e.g. L 390-395 HOA considered as a mixture of traffic and biomass burning. Could it be that instead of BB, HOA could be considered as more of a mixture between traffic and combustion from central heating units? + (Technical corrections) L394-395: Rephrase

In the present work, the HOA factor is a mixed factor with traffic POA and biomass burning POA in winter. The latter mixing agent could be linked to residential wood burning emissions due to heating purposes during cold months in Pairs. The HOA factor has been further explained linking to possible source sectors. It now reads as:

*"…However, HOA evening peaks occurs globally later than eBC$_{ff}$ and NO$_x$ ones (9:00-10:00 PM vs. 7:00 PM, respectively) and much lower ratios are observed between HOA and eBC$_{ff}$ in the morning than in the evening. This might be partly explained by i) higher eBC traffic emission factor in the morning and/or ii) impacts of residential heating sources, e.g., wood and/or heating oil burning (Lin et al., 2018) on the HOA concentrations in the late evening.*

*Moreover, eBC$_{ff}$ shows a clear weekend effect, with less-pronounced pattern on Saturday and Sunday due to road transport reduction, while HOA displays intense nighttime peaks during weekend. This HOA mean pattern was substantially influenced by winter data, whereas summertime patterns display better consistency between HOA, eBC$_{ff}$ and NO$_x$ (Figure S12).* *Altogether, these results suggest that this HOA is considered as a mixed factor partly composed of both, traffic and residential heating aerosols.* *This statement is in good agreement with conclusions from complementary studies showing wood burning contribution to HOA at the same site (Petit et al., 2014; Srivastava et al., 2019)."*

**Technical corrections:**

L488-489 More recent studies also report part of the low-volatility (more oxidized) OOA originating from primary combustion sources (e.g. Stavroulas et al., 2019).

Thanks for the comments. This reference has been cited to support the origins of MO-OOA partly linking to biomass burning source. It now reads as:

"*... As shown in Figure S15, the correlations between MO-OOA and sulfate are found to be strongly BBOA- and wind speed-dependent. For high wind speed and low BBOA concentrations, the mean MO-OOA-to-sulfate ratio is close to 1, while it reaches up to 8 under high BBOA and low-to-medium wind speed. This is consistent with the assumption of an enhancement of MO-OOA formation in the presence of substantial biomass burning emissions, which have been reported as a major anthropogenic SOA source (Heringa et al., 2011; Tiitta et al., 2016; Bertrand et al., 2017;* *Stavroulas et al., 2019;* *Daellenbach et al., 2019)* *Actually, both MO-OOA and LO-OOA factors may be significantly influenced by wood burning emissions as they are displaying similar correlations with eBC$_{wb}$ for highest MO-OOA-to-sulfate ratios during wintertime (Figure 6).*"

**References:**

Cubison, M. J., Ortega, A. M., Hayes, P. L., Farmer, D. K., Day, D., Lechner, M. J., Brune, W. H., Apel, E., Diskin, G. S., Fisher, J. A., Fuelberg, H. E., Hecobian, A., Knapp, D. J., Mikoviny, T., Riemer, D., Sachse, G. W., Sessions, W., Weber, R. J., Weinheimer, A. J., Wisthaler, A.,

and Jimenez, J. L.: Effects of aging on organic aerosol from open biomass burning smoke in aircraft and laboratory studies, Atmos. Chem. Phys., 11, 12049-12064, 10.5194/acp-11-12049-2011, 2011.

Fröhlich, R., Cubison, M. J., Slowik, J. G., Bukowiecki, N., Canonaco, F., Croteau, P. L., Gysel, M., Henne, S., Herrmann, E., Jayne, J. T., Steinbacher, M., Worsnop, D. R., Baltensperger, U., and Prévôt, A. S. H.: Fourteen months of on-line measurements of the non-refractory submicron aerosol at the Jungfraujoch (3580 m a.s.l.) – chemical composition, origins and organic aerosol sources, Atmos. Chem. Phys., 15, 11373-11398, 10.5194/acp-15-11373-2015, 2015.

Lin, C., Huang, R.-J., Ceburnis, D., Buckley, P., Preissler, J., Wenger, J., Rinaldi, M., Facchini, M. C., O'Dowd, C., and Ovadnevaite, J.: Extreme air pollution from residential solid fuel burning, Nature Sustainability, 1, 512-517, 10.1038/s41893-018-0125-x, 2018.

Middlebrook, A. M., Bahreini, R., Jimenez, J. L., and Canagaratna, M. R.: Evaluation of Composition-Dependent Collection Efficiencies for the Aerodyne Aerosol Mass Spectrometer using Field Data, Aerosol Sci. Technol., 46, 258–271, https://doi.org/10.1080/02786826.2011.620041, 2012.

Rattanavaraha, W., Canagaratna, M. R., Budisulistiorini, S. H., Croteau, P. L., Baumann, K., Canonaco, F., Prevot, A. S. H., Edgerton, E. S., Zhang, Z., Jayne, J. T., Worsnop, D. R., Gold, A., Shaw, S. L., and Surratt, J. D.: Source apportionment of submicron organic aerosol collected from Atlanta, Georgia, during 2014–2015 using the aerosol chemical speciation monitor (ACSM), Atmospheric Environment, 167, 389-402, https://doi.org/10.1016/j.atmosenv.2017.07.055, 2017.

Srivastava, D., Daellenbach, K.R., Zhang, Y., Bonnaire, N., Chazeau, B., Perraudin, E., Gros, V., Villenave, E., Prévôt, A.S.H., El Haddad, I., Favez, O., and Albinet, A.: Comparison of different methodologies to discriminate between primary and secondary organic aerosols, Sci. Total Environ., 690, 944-955, 2019.

Stavroulas, I., Bougiatioti, A., Grivas, G., Paraskevopoulou, D., Tsagkaraki, M., Zarmpas, P., Liakakou, E., Gerasopoulos, E., and Mihalopoulos, N.: Sources and processes that control the submicron organic aerosol composition in an urban Mediterranean

environment (Athens): a high temporal-resolution chemical composition measurement study, Atmos. Chem. Phys., 19, 901-919, 10.5194/acp-19-901-2019, 2019.

Ulbrich, I. M., Canagaratna, M. R., Zhang, Q., Worsnop, D. R., and Jimenez, J. L.: Interpretation of organic components from Positive Matrix Factorization of aerosol mass spectrometric data, Atmos. Chem. Phys., 9, 2891-2918, 10.5194/acp-9-2891-2009, 2009.

van Marle, M. J. E., Kloster, S., Magi, B. I., Marlon, J. R., Daniau, A. L., Field, R. D., Arneth, A., Forrest, M., Hantson, S., Kehrwald, N. M., Knorr, W., Lasslop, G., Li, F., Mangeon, S., Yue, C., Kaiser, J. W., and van der Werf, G. R.: Historic global biomass burning emissions for CMIP6 (BB4CMIP) based on merging satellite observations with proxies and fire models (1750–2015), Geosci. Model Dev., 10, 3329-3357, 10.5194/gmd-10-3329-2017, 2017.

Zhang, Y. J., Tang, L. L., Wang, Z., Yu, H. X., Sun, Y. L., Liu, D., Qin, W., Canonaco, F., Prévôt, A. S. H., Zhang, H. L., and Zhou, H. C.: Insights into characteristics, sources, and evolution of submicron aerosols during harvest seasons in the Yangtze River delta region, China, Atmos. Chem. Phys., 15, 1331-1349, 10.5194/acp-15-1331-2015, 2015.

Zotter, P., Herich, H., Gysel, M., El-Haddad, I., Zhang, Y., Močnik, G., Hüglin, C., Baltensperger, U., Szidat, S., and Prévôt, A. S. H.: Evaluation of the absorption Ångström exponents for traffic and wood burning in the Aethalometer-based source apportionment using radiocarbon measurements of ambient aerosol, Atmos. Chem. Phys., 17, 4229-4249, 10.5194/acp-17-4229-2017, 2017.

---

## Author Response (AR2)

Dear Dr. Eleanor Browne,

We sincerely thank you very much for your comments and time. We have revised the manuscript accordingly. The corresponding modification in the revised manuscript is highlighted in red.

Sincerely yours,

On behalf of the authors,

Yunjiang Zhang and Olivier Favez

Line 90: Add "being" after "based on"

Added.

Line 173: Please state (as is done in the response to referees) that 0.9 and 1.7 were estimated by the AAE distribution observed in the study.

It has been addressed in the revised manuscript as follows:

"For these calculations, absorption Angström exponent (AAE) values - in the wavelength range 470-950 nm - of 0.9 and 1.7 for eBC$_{ff}$ and eBC$_{wb}$, respectively, were estimated by the AAE distributions observed in this study (see Figure S2)."

[Figure]

**Figure S2.** Frequency distributions of ambient absorption Angström exponent (AAE) values calculated at different paired-wavelengths (i.e., 370 and 950 nm, as well as 470 and 950 nm) and for the 6 wavelengths comprised between 470 and 950 nm.

Line 418: This should be Figure S13 not S12

Thanks for the correction. It has been revised.